# Towards a swath-to-swath sea-ice drift product for the Copernicus Imaging Microwave Radiometer mission.

Thomas Lavergne[1], Montserrat Piñol Solé[2], Emily Down[1], Craig Donlon[2]

[1]Research and Development Department, Norwegian Meteorological Institute, Oslo, Norway
[2]European Space Agency, Keplerlaan 1, 2201AZ Noordwijk, the Netherlands

*Correspondence to*: Thomas Lavergne (thomas.lavergne@met.no)

**Abstract.**

Across spatial and temporal scales, sea-ice motion has  implications on ship navigation, the sea-ice thickness distribution, sea ice export to lower latitudes and re-circulation in the polar seas, among others. Satellite remote sensing is an effective way to monitor sea-ice drift globally and daily, especially using the wide swaths of passive microwave missions. Since the late 1990s, many algorithms and products have been developed for this task. Here, we investigate how processing sea-ice drift vectors from the intersection of individual swaths of the Advanced Microwave Scanning Radiometer 2 (AMSR2) mission compares to today's status-quo (processing from daily averaged maps of brightness temperature). We document that the "swath-to-swath" (S2S) approach results in many more (two orders of magnitude) sea-ice drift vectors than the "daily-maps" (DM) approach. These S2S vectors also validate better when compared to trajectories of on-ice drifters. For example, the RMSE of the 24 hour winter Arctic sea-ice drift is 0.9 km for S2S vectors, and 1.3 km for DM vectors from the 36.5 GHz imagery of AMSR2. Through a series of experiments with actual AMSR2 data and simulated Copernicus Imaging Microwave Radiometer (CIMR) data, we study the impact that geo-location uncertainty and imaging resolution have on the accuracy of the sea-ice drift vectors. We conclude by recommending that a "swath-to-swath" approach is adopted for the future operational Level-2 sea-ice drift product of the CIMR mission. We outline some potential next steps towards further improving the algorithms, and making the user community ready to fully take advantage of such a product.

## 1    Introduction

The balance between air drag, ocean drag, and lateral forces control the motion of sea ice (Leppäranta, 2005). At the local scale, sea-ice motion can both be a facilitator and impediment to ship navigation, opening and closing routes, opening leads or forming pressure ridges. At the larger regional to basin scales, sea-ice motion (aka sea-ice drift) exports sea-ice to lower latitudes where it melts, contributing to the redistribution of fresh water. Inside the Arctic Ocean, re-circulation of sea-ice (e.g. in the Beaufort Sea) leads to the ageing and thickening of the ice pack towards the northern coasts of the Canadian Arctic Archipelago and Greenland (Timmermans and Marshall, 2020). Sea-ice drift also plays a role in sea-ice formation and ocean circulation via the formation of coastal latent heat polynyas (Ohshima et al., 2016), as well as in the transport of

sediments and other tracers across ocean basins (Krumpen et al., 2019). With climate change, the area and thickness of sea-ice is reduced in the Arctic, which leads to a more mobile sea-ice cover, and positive trends in sea-ice velocity (Spreen et al. 2011; Kwok et al. 2013). Trends in sea-ice motion, linked to trends in wind speed, are also observed in the Southern
Hemisphere (Holland and Kwok, 2012; Kwok et al. 2017).

In the Arctic, on-ice buoys are regularly deployed. Using various satellite-based positioning and communication technologies, they record and report their position at regular intervals that can be stacked into trajectories. These on-ice buoys can inform on the general patterns of sea-ice motion and their response to atmospheric circulation (Rigor et al. 2002), but also the hourly to sub-hourly patterns like the effects of tides and inertial oscillations (Mc Phee, 1978; Heil and Hibler,
2002). Thanks to a sustained and coordinated deployment program (e.g. the International Arctic Buoy Programme, IABP), on-ice buoys also inform on the climate trends (Rampal et al. 2009). Finally, on-ice buoys are critical to the validation and tuning of model-simulated (e.g. Schweiger and Zhang, 2015; Rampal et al. 2016) and satellite-derived (e.g. Kwok et al., 1998; Lavergne et al. 2010; Sumata et al., 2014) sea-ice drift information. Nevertheless, buoys have a limited lifespan before the sea-ice floe they seat on melts, or they drift out of the Arctic, or they suffer technical issues; this and limited
opportunities for deployment result in sparse spatial coverage. This is even more true in the Southern Hemisphere where the annual cycles of sea-ice cover, and fewer research cruises, strongly limit the availability of on-ice platforms.

Consequently, satellite remote sensing has developed as an attractive option to monitor sea-ice drift consistently across the polar sea-ice cover, at a daily to sub-daily frequency. The initial work by Ninnis et al. (1986) was followed by many investigators, using a variety of satellite imaging sensor technologies as input, including visible/infrared radiometry (Emery
et al. 1991), microwave radiometry and scatterometry (Agnew et al. 1997; Kwok et al. 1998; Liu et al. 1999; Lavergne et al. 2010; Girard-Ardhuin and Ezraty, 2012), and Synthetic Aperture Radar (SAR) (Kwok et al. 1990; Komarov and Barber, 2013; Muckenhuber et al. 2016). The various imaging technologies however lead to sea-ice motion fields with different characteristics, e.g. medium spatial resolution (~20 km) and coverage limited by cloud cover for the visible/infrared radiometry, high spatial resolution (~5-10 km) but coverage limited by acquisition repeat cycles for the SAR imagery, and
coarse spatial resolution (> 30 km) and daily complete coverage for the microwave radiometers and scatterometers. Despite the imaging technologies being very different from each other, the motion tracking algorithms employed are quite similar, and stem from the Maximum Cross Correlation (MCC) technique (Emery et al. 1986).

There is however one trait of these various sea-ice drift products that is quite different between visible/infrared radiometry and SARs on the one side, and microwave radiometry and scatterometry on the other side. The former are always computed
from the overlap of two individual swaths (or scenes), while the latter is traditionally computed from daily averaged maps of the satellite signal (brightness temperature or backscatter coefficient). Working with daily averaged maps is more straightforward since one does not have to deal with the borders of the individual swaths. But it is also intuitively not optimal, since the very motion under study could blur the aggregated satellite image and lead to a product with poorer accuracy. Still, processing from daily averaged satellite images has so far been the norm for this class of coarser resolution
sensors. Maslanik et al. (1998) note that *preliminary results are mentioned that show use of individual swath data with*

*radiometric correction and better geolocation does not significantly improve the comparison with buoys in the Arctic* [*with respect to using daily maps*, our addition] and identify as future work *to document whether orbital data, rather than the 24-hour averaged TBs used by each group, offers significant improvements*.

Here, we report on such a study that was conducted in the context of the design phase for a future satellite mission: the Copernicus Imaging Microwave Radiometer (CIMR). CIMR is a conically-scanning microwave radiometer mission under study at the European Space Agency for the Expansion phase (2026-2030) of the European Union's Copernicus Space Component. At time of writing, the reference document for the CIMR mission is the Mission Requirement Document (Donlon et al., 2020).

The aim of our research is thus two-fold: 1) to document the pros and cons of processing sea-ice drift vectors from individual orbits vs from daily averaged maps, and 2) to discuss the implications for future sea-ice motion capabilities of the CIMR mission. This paper is structured as follows: satellite and in-situ data are introduced in Sect. 2, while the methodologies for sea-ice motion tracking and product validation are covered in Sect. 3. Section 4 documents our results, Sect. 5 covers a discussion in the context of the CIMR mission, and we conclude in Sect. 6.

## 2    Data

### 2.1    Orbit-based brightness temperature data

We accessed Level-1b brightness temperature ($T_B$) data (Version 2 calibration) of the Global Change Observation Mission 1st-Water (GCOM-W1) Advanced Microwave Scanning Radiometer 2 (AMSR2) directly from the Japan Aerospace Exploration Agency (JAXA) Global Portal System ([https://gportal.jaxa.jp/gpr/](https://gportal.jaxa.jp/gpr/), last accessed 26[th] february 2021). For this study, we used the brightness temperatures at both Vertical and Horizontal polarizations of the Ka (36.5 GHz) and W (89 GHz) imagery channels. Table 1 summarizes the spatial resolution of the microwave imagery channels of the AMSR2 and CIMR missions.

| Band | L | C | X | Ku | Ka | W |
|---|---|---|---|---|---|---|
| Center Frequency [GHz] | 1.4 | 6.9 | 10.7 | 18.7 | 36.5 | 89.0 |
| AMSR2 [km] | - | 49 | 33 | 18 | 9 | 4 |
| CIMR [km] | <60 | 15 | 15 | 5 | <5 | - |

**Table 1:  Spatial resolution (computed as the arithmetic mean of the minor and major diameters of the instantaneous field-of-view ellipse) of selected microwave frequencies of the AMSR2 and CIMR missions. AMSR2 also records at 7.3 and a 23.8 GHz, those will not be on-board CIMR. "-" indicates a microwave frequency is not recorded by the mission. The values for CIMR are from Donlon et al. (2020), those for AMSR2 from the Observing Systems Capability Analysis and Review (OSCAR) tool of the World Meteorological Organization. See also Lavergne (2018) for a graphical representation of these values.**

We use AMSR2 data from two periods: from 1$^{st}$ October 2019 to 31$^{st}$ December 2020 for the Northern Hemisphere (15 months), and from 1$^{st}$ June to 31$^{st}$ August 2016 for the Southern Hemisphere (3 months). These two periods are selected because they include winter freezing conditions in both regions and the summer melt season in the Northern Hemisphere, as well as to maximize the number of available on-ice drifters available for validation, as covered in the next section.

## 2.2    GPS trajectories of on-ice drifters

To validate satellite-based sea-ice drift vectors, we access GPS trajectories for on-ice drifters in the Arctic and Antarctic (Fig. 1). In the Arctic, buoys are in the Beaufort Gyre, and in the Transpolar Drift. In the Antarctic, all buoys are in the central Weddell Sea. The colors in Fig. 1 represent the time of the observation records within the two periods.

A variety of buoy types are included in our validation data, but we are only concerned with three pieces of information per trajectory record: timestamp, latitude, and longitude. Most buoys report positions on an hourly basis. The Ice-Tethered Profiler data were collected and made available by the Ice-Tethered Profiler Program (Toole et al., 2010; Krishfield et al., 2008) based at the Woods Hole Oceanographic Institution (http://www.whoi.edu/itp, last accessed 1$^{st}$ June 2020). A variety of other buoys were accessed from the data portal meereisportal.de (Grosfeld et al., 2016, last accessed 1$^{st}$ June 2020), including all Antarctic buoys, and the buoys deployed at and around the Multidisciplinary drifting Observatory for the Study of Arctic Climate (MOSAiC) site (Krumpen et al. 2020).

## 2.3    Sea Ice Concentration data from EUMETSAT OSI SAF

The Sea Ice Concentration (SIC) data from  European Organization for the Exploitation of Meteorological Satellites (EUMETSAT) Ocean and Sea Ice Satellite Application Facility (OSI SAF) was accessed and transformed into an ice/no-ice mask using a threshold of 40% SIC: grid cells with a SIC above 40% are considered as ice-covered, and can be used for the computation of sea-ice drift vectors. The Interim Climate Data Record "v2" based on Special Sensor Microwave Imager Sounder (SSMIS) data is used (OSI-430-b). Algorithms and processing chains are described in Lavergne et al. (2019).

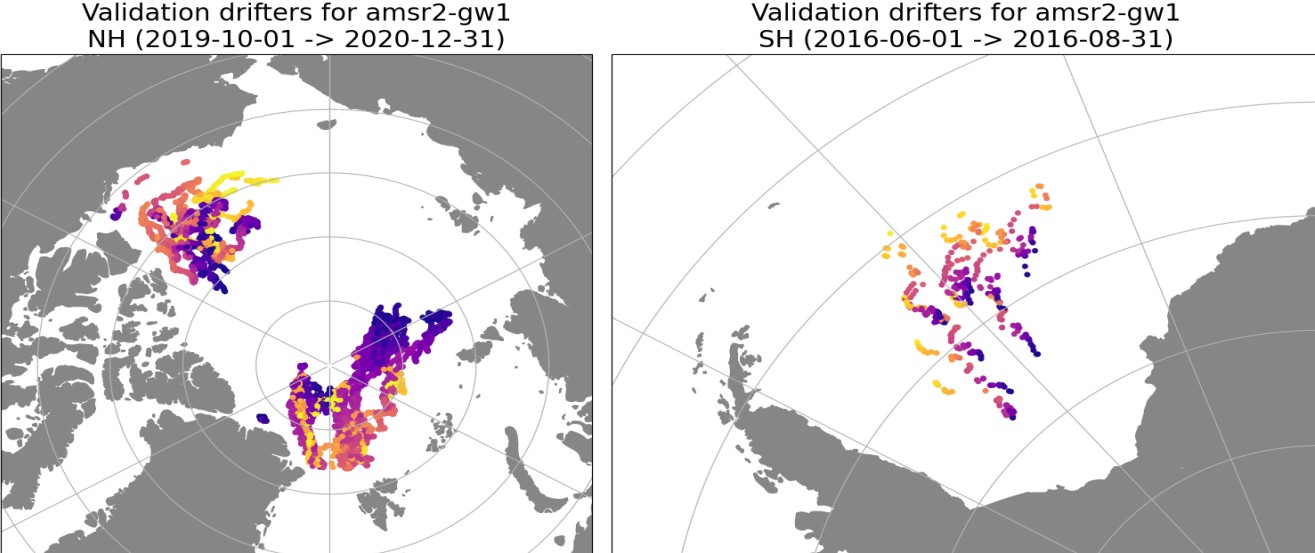

**Figure 1: Location of on-ice drifter trajectories accessed for validation of the sea-ice drift vectors in the Northern Hemisphere (left) and Southern Hemisphere (right). The colors represent the time along the trajectory within the 15-months period (NH) and 3-month periods (SH).**

## 3 Methodology

### 3.1 Sea-ice motion tracking

The sea-ice motion tracking methodology implemented here, including the quality control steps, is very similar to that described by Lavergne et al. (2010) and implemented in the operational chains of the EUMETSAT OSI SAF (Lavergne et al., 2016). We recall below three unique features of this processing chain.

First, it implements the Continuous Maximum Cross-Correlation (CMCC) motion tracking algorithm. The CMCC stems from the well-known MCC (Ninnis et al. 1986, Emery et al. 1986) but implements a continuous optimization of the cross-correlation function (rather than a discrete optimization in MCC). The continuity is enabled by on-the-fly linear interpolation of image pixels. This continuous optimization strongly reduces the "quantization noise" present in many MCC-based sea-ice drift products. Lavergne et al. (2010) documented how the CMCC-based ice motion vectors were more accurate than those based on MCC (see also Hwang 2013, Sumata et al. 2014).

Second, for a given microwave frequency, the information content of both the vertically and horizontally polarized images are combined within the optimization of the cross-correlation function. In practice, and following Lavergne et al. (2010), the

solution sea-ice drift vectors are at the maximum of the sum of two cross-correlation functions : one from of the vertically polarized imagery, and one from the horizontally polarized imagery. The reader is referred to the discussion in Lavergne et al. (2010, Sect. 3.2) for a discussion of this approach. In the remaining of our paper, despite mentionning only the microwave frequency, we do use both polarizations in the motion tracking.

    Third, it implements an iterative quality control step to detect and correct a few questionable (*aka* "rogue") vectors. Those
vectors are at the maximum of the cross-correlation function, but point in a direction completely different from the neighbouring vectors. All block-based motion tracking algorithms need such a quality control step (e.g. Girard-Ardhuin and Ezraty, 2012; Haarpaintner, 2006; Tschudi et al. 2020), but most authors remove the rogue vectors and the vector field has missing data cells. The quality control step of Lavergne et al. (2010) both detects the questionable vectors and -most of the time- corrects them, reducing the occurrence of data gaps.

As in Lavergne et al. (2010) and in the EUMETSAT OSI SAF sea-ice drift product (Lavergne et al. 2016), we process sea-ice drift vectors with a grid spacing of 62.5 km on two polar stereographic grids (Arctic and Antarctic).

### 3.2    "swath-to-swath" and "daily maps" sea-ice drift products

    The sea-ice drift product run at the EUMETSAT OSI SAF based on the algorithms of Lavergne et al. (2010) is processed from daily gridded maps of satellite signal: daily averaged maps of $T_B$ from the JAXA AMSR2 and United States
Department of Defence (DoD) SSMIS, and daily averaged backscatter coefficients (corrected to 40° incidence angle) of the EUMETSAT Advanced SCATterometer (ASCAT). These daily averaged maps aggregate satellite signal from 00 UTC to 23:59 UTC for a given day, and their valid time is 12 UTC. We name this type of sea-ice drift product a "daily maps" product (noted DM).

    It is noteworthy that, in addition to averaged maps of satellite signal from which DM sea-ice drift vectors are computed, we
prepare the corresponding maps of daily averaged satellite sensing time: the average of the observation time from all the swaths during the 24 hours period of the daily averaging (Fig. 2 in Lavergne et al. 2016). This mean sensing time is an important information attached to each DM sea-ice drift vector: even though they are from daily maps with valid time 12 UTC, the DM vectors will be used and validated taking into account these space varying mean start and stop times.

    For this study, we prepared both DM products and "swath-to-swath" products (noted S2S). S2S sea-ice drift products are
processed from swaths of satellite signal that are gridded individually, thus without daily averaging. For a given day, there are typically 12 to 14 such individual gridded orbits per satellite sensor, each having a different valid time separated by approximately 100 minutes. The sea-ice area at the overlap of two such individually gridded maps is where S2S drift vectors

are computed (see Fig. 2). When preparing the individually gridded swaths of brightness temperatures, we also prepare the corresponding grids of sensing time for later use in the validation.

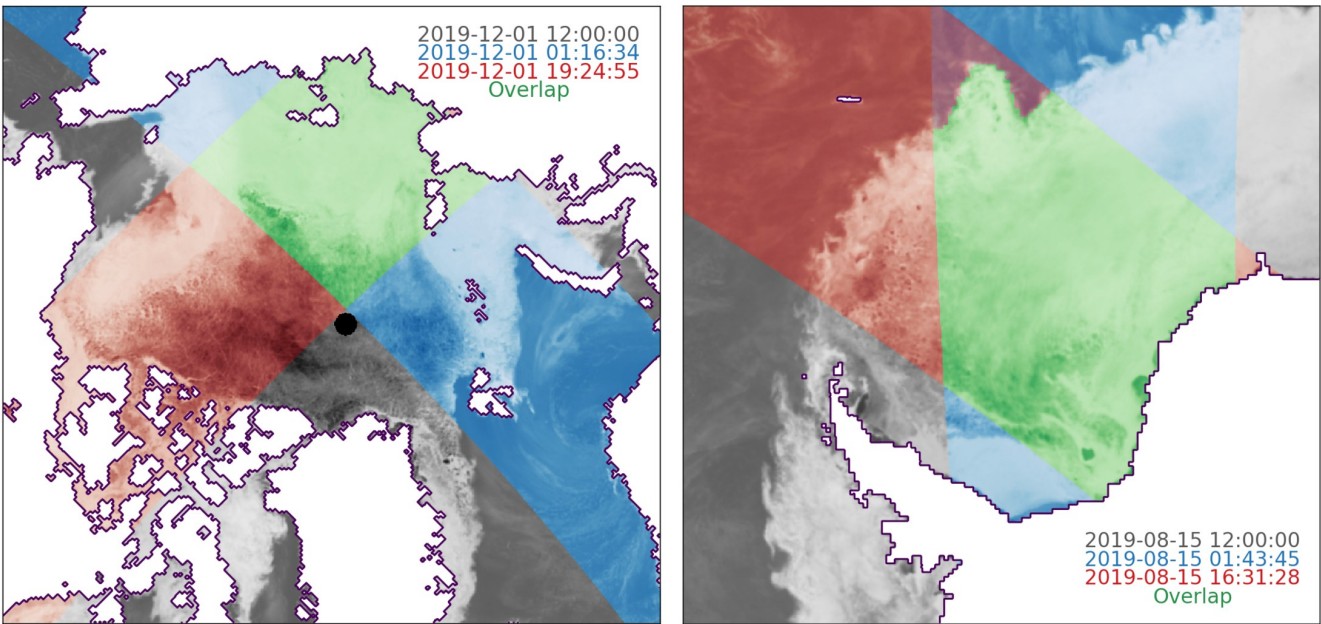

2019-12-01 12:00:00
2019-12-01 01:16:34
2019-12-01 19:24:55
Overlap

2019-08-15 12:00:00
2019-08-15 01:43:45
2019-08-15 16:31:28
Overlap

**Figure 2: Left: daily average map of AMSR2 36.5 GHz V-pol T$_B$ on 1 December 2019 (greys) for the Arctic, and two individual gridded swaths on the same day (blues: 01:16:34 UTC, reds: 19:24:55 UTC). The sea-ice region of overlap between the two swaths is highlighted in greens, and is where S2S drift vectors can be computed. Right: similar but for the Antarctic on 15 August 2019 (blues: 01:43:45 UTC, reds: 16:31:28 UTC)**

We prepare two periods of S2S and DM ice drift products from GCOM-W1 AMSR2 36.5 GHz T$_B$ imagery: from 1$^{st}$ October 2019 to 31$^{st}$ December 2020 for the Northern Hemisphere, and from 1$^{st}$ June to 31$^{st}$ August 2016 for the Southern Hemisphere (see Sect. 2.1). In both periods, we process sea-ice drift vectors with durations (*aka* time-span) ranging from ~100 minutes (the separation time between two consecutive orbits) to 48 hours. The duration of an ice drift vector is the difference between the timestamps of the start and stop image from which motion is estimated (these images are either individual swaths for S2S products or daily maps of DM products).

### 3.3    Collocation of satellite and in-situ drift vectors

Comparison between any two data sets first requires deciding on a collocation strategy, in space and time. To collocate sea-ice motion vectors is somewhat different from collocating other geophysical variables, in that the reference data source (a timeseries of GPS records from the ice drifting buoy or ship) is not directly comparable to what the satellite product measures (a net Lagrangian displacement vector over a time period). For this validation exercise, we follow the approach of

Lavergne et al. (2010), Hwang (2013) and others to compute equivalent net Lagrangian displacement vectors from the in-situ trajectories as part of the collocation step. In short, for each satellite vector to be validated, the in-situ record closest to the start position of the satellite vector is selected and -if within a suitable geographical radius- the in-situ net Lagrangian displacement vector is computed from its GPS records, using the start and end time of the satellite vector. We rely on a nearest neighbor approach in both space and time domains (nearest position, nearest time) without interpolation.

For a collocation pair to enter in the collocated matchup database, the start time of the in-situ displacement has to be within plus or minus 3 hours of the start time of the nearest satellite drift vector, the duration of the in-situ vector has to be within plus or minus 1 hour of that of the satellite vector, and the distance between the start positions of the in-situ and satellite vectors has to be less than 30 km. To avoid over-representation in the case of buoy clustering (e.g. buoy arrays or the MOSAiC site) only the closest buoy to a satellite vector is kept. In addition, the collocated matchup database is filtered so
that no directly neighbouring satellite vectors co-exist in it, in order to reduce the effects of the correlation lengths stemming from the satellite retrieval.

For the S2S product, the start and end time of the drift vectors are the time stamp of the individual satellite swaths. For the DM products, they are the space-varying mean observation (overflight) times at the start and end images, meaning that the satellite product is not used as if starting and stopping at 12 UTC everywhere in the product grid.

**3.4    Simulation of CIMR orbits and swaths**

The swath of CIMR (>1900 km) will be larger than that of AMSR2 (1450 km). To study the impact of a wider swath on the characteristics of a future S2S sea-ice drift product from CIMR, we simulate some of its orbits and swaths. CIMR is to fly along a sun-synchronous dawn-dusk orbit with a 98.7° inclination. Additional orbit and instrument parameters relevant to the simulations are in Donlon et al. (2020).

We simulate two consecutive days of CIMR orbit and swath coverage using an ad-hoc tool relying on the Earth Observation Mission Customer Furnished Item SoftWare (EO CFI SW) libraries (http://eop-cfi.esa.int/index.php/mission-cfi-software/eocfi-software, last accessed 23rd October 2020). In practice, the orbit propagation is performed using the mean keplerian orbit propagation mode available in the orbit library. The spacecraft attitude is modelled applying local normal pointing and yaw steering law, and the instrument swath edges are defined in
terms of direction look angles. Finally the zone visibility functions were used to compute the coverage mask of the instrument swath over the Northern and Southern Hemisphere grids separately. The visibility library includes functions to calculate the intersection of the swath points with the Earth ellipsoid, which are invoked internally by the zone visibility routine.

The coverage mask (binary map) of individual swath is then used to mask a daily-averaged map of AMSR2 TB to create a
simulated CIMR TB swath, that enters the sea-ice motion software. Our simulated CIMR swaths are thus only for studying
the impact of the wider swaths, not the better spatial resolution.

## 4    Results

### 4.1    Comparative space/time coverage of the DM and S2S vectors

Given the full daily coverage of GCOM-W1 AMSR2 (except the observation gap at the pole in the Arctic), the number of
potential DM sea-ice drift vectors is a function of the sea-ice extent only. Conversely, the number of potential S2S sea-ice
drift vectors varies with sea-ice extent and with the area of overlap between the individual swaths. This area of overlap
depends on the relative orientation of two swaths, thus on the time difference between the orbits. Consecutive orbits will
allow a larger overlap than, e.g. orbits that are 4-8 hours apart. Due to the orbit configuration of the GCOM-W1 mission, two
orbits that are roughly 24 hours apart will have large overlaps.

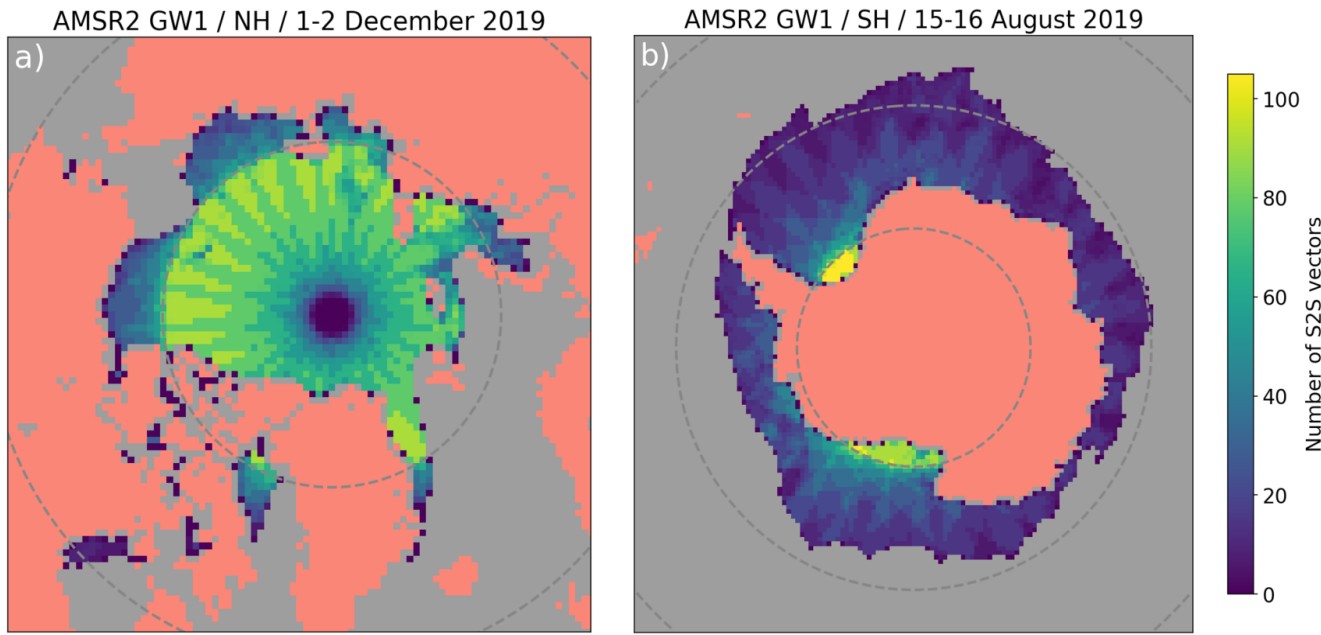

**Figure 3: a) number of S2S vectors per grid cell in the Northern Hemisphere for the period 1-2 December 2019 and GCOM-W1
AMSR2 mission. b) same quantity but for the Southern Hemisphere and for the period 15-16 August 2019. Parallels at +/- 75 and
+/-60 are drawn.**

Fig. 3 shows the spatial distribution of the number of S2S drift vectors that start and/or stop in the period 1-2 December 2019
in the Arctic (left) and in the period 15-16 August 2019 in the Antarctic (right). The latitude dependency is clearly visible in

both the Northern Hemisphere (NH) and Southern Hemisphere (SH) grids (the parallels at +/-75°, +/-60° are drawn). For the AMSR2 instrument, the regions poleward of +/-75° correspond to the areas with most inter-swath overlap. In total 110,441 S2S ice drift vectors are computed in the period 1-2 December 2019 in NH, while only 1,729 DM vectors are available from the same two days. Similarly, 74,799 S2S ice drift vectors are computed in the period 15-16 August 2019 in SH, but only 3,617 DM vectors are available from the same two days. The sea-ice extent in NH on 1 December 2019 was 10.8

million $km^2$ and 17,9 millions $km^2$ in SH on August 15[th] (according to the OSI SAF "v2" SIC data, see Sect. 2.3). This difference is well reflected in the larger number of DM vectors computed in SH than NH. Conversely, the higher number of S2S vectors in NH than SH is controlled by the different latitudinal distribution of the sea-ice cover as illustrated on Fig. 3. In NH, the majority of S2S vectors are computed in a band between 75°N and 80°N (up to 90 S2S drift vectors per grid cell), and slowly decays towards 85°N with spatial patterns that are typical of satellite swath geometry. The sharp decay poleward

of 85°N is due to the polar observation hole of the AMSR2 instrument, with a width of ~ 0.5° (see Fig. 2, left panel). Two factors amplify this decay. First, there is less overlap between swaths at very high latitude (edges of the swaths). Second, the motion tracking algorithm works with sub-windows (e.g. 11x11 image pixels) and this limits the retrieval of drift vectors in the immediate vicinity of areas with image missing data (e.g. the polar observation hole, sea-ice edge, coastal region, etc…). In fact not a single S2S vector (nor DM vector) are retrieved north for 88.5°N from the GCOM-W1 AMSR2 data. The

transition from ~90 S2S drift vectors per grid cell to 40 and less equatorward is visible on both NH and SH maps. In SH, most of the sea-ice cover is south of 75°S, which is the case all year round. Still, even at the outskirts of the SH sea-ice cover, we have 5-15 S2S vectors - compared to a single DM vector - per grid cell. The numbers above pertain to the configuration where drift vectors are processed at a grid spacing of 62.5 km: reducing the grid spacing would directly increase the number of DM and S2S vectors.

The first advantage of adopting a S2S ice drift processing for microwave radiometer satellite data is thus to take full advantage of the individual swaths, and get access to many more sea-ice drift vectors than in the DM configuration.

    These S2S sea-ice drift vectors however have very different characteristics compared to the DM products, especially in the time and duration domain. The histograms in Fig. 4 show the distribution of the S2S and DM vectors from the period 1-2 December 2019 (NH, top row) and 15-16 August 2019 (SH, bottom row) in terms of start time of the drift (left), end time of

the drift (middle), and duration (right). We first note that the distributions of DM vectors is not a single value at 12 hours (start), 36 hours (stop) and 24 hours (drift duration), but that we report some variation around those values. As introduced earlier, the start and stop times of DM vectors are the space-varying mean observation time of all the swaths during the 24 hour period of the daily averaging. Despite the variations, the distributions of start time, stop time, and duration are concentrated around expected peak values at 12 hours, 36 hours, and 24 hours, respectively. Noticeably, the duration of most

DM vectors falls within plus or minus 2 hours of 24 hours. Nevertheless, these mean times associated with the DM ice drift vectors are values averaged over several overlapping swaths, and sometimes do not provide a faithful representation of the

time characteristics of the ice drift product, e.g. when a region of the sea-ice cover is observed twice in the early morning, and once in late afternoon.

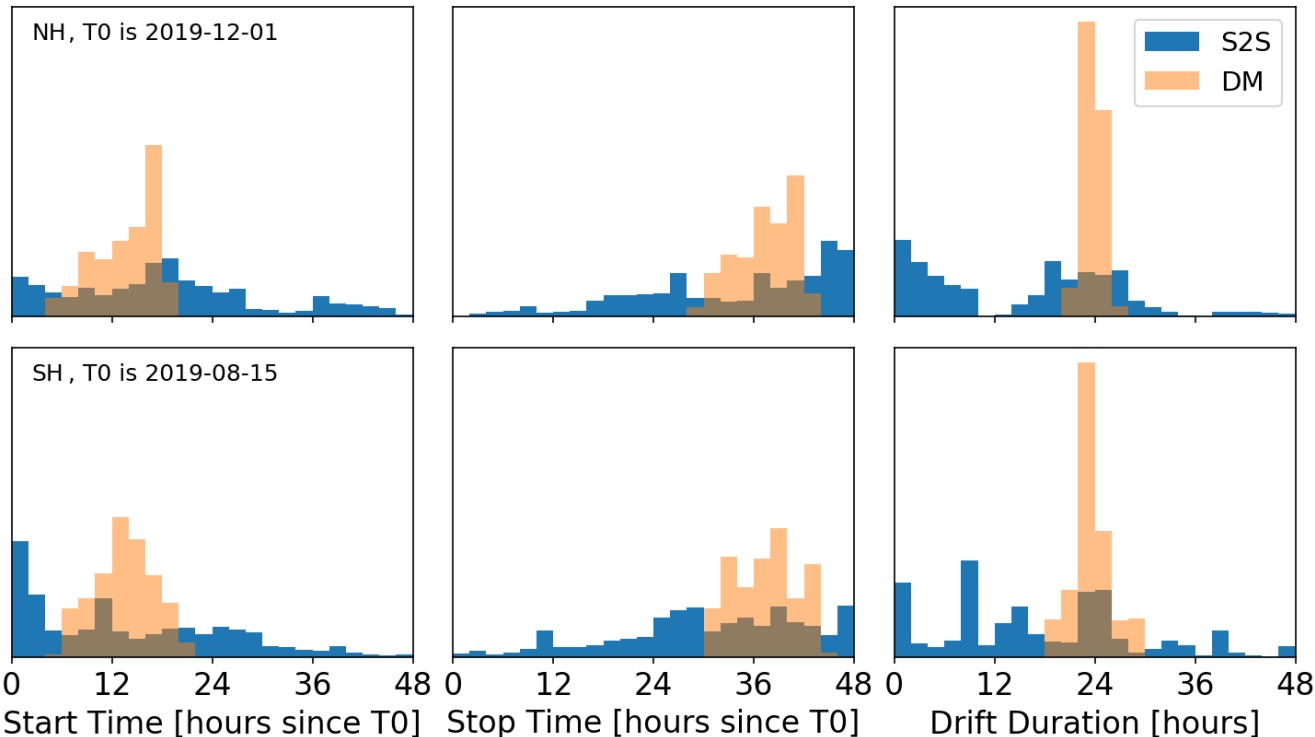

**Figure 4: Distribution of S2S (blue) and DM (orange) vectors derived from GCOM-W1 AMSR2 imagery in Northern Hemisphere (top row) and Southern Hemisphere (bottom row): Start time (left), end time (middle), and duration of the drift vectors (right).**

Conversely, the temporal information attached to S2S drift vectors is much more accurate, and has significantly different distributions in Fig. 4 when compared to those of DM. The distribution of start time (left) and stop time (middle) covers the whole range in the 48 hours period, with peaks around 0, 18, and 40 hours past 1 December at 00 UTC (start time) and 25 and 48 hours past 1st December at 00 UTC (stop time). These peaks correspond to an increased number of S2S drift vectors being available at the end of the 48 hours period, and is thus a function of many parameters including the orbit and swath characteristics of the satellite mission (in our case GCOM-W1 AMSR2), the extent and geographic repartition of the sea-ice cover, both of which combine into overlap characteristics. The S2S histograms in Fig. 4 should thus be seen as an illustration of a general pattern (repartition over the whole period, with peaks). The S2S duration histograms (Fig. 4, right panels) are also controlled by the characteristics of the orbit, swath width and the sea-ice cover. They document that a wide spectrum of

drift durations are recorded by S2S drift products, with peaks near 0, 24, and 48 hours in the NH and more spread in the SH. These peaks generally correspond to when the swaths overlap most.

All in all, the short analysis conducted here documents that S2S and DM ice-drift products have distinct characteristics. S2S ice-drift products offer many more vectors, and this number varies with latitude. The two types of products cover the temporal domain differently. The S2S products have a broad but not homogeneous coverage in start times, stop times, and duration. Our next step is to investigate if the accuracy of the S2S ice-drift product is better, similar, or worse than that of DM ice-drift products. We assess this accuracy against collocated in-situ drifter GPS trajectories. Results are reported in the next section.

## 4.2    Validation of 24 hour drift vectors against buoy data

We collocate all S2S and DM sea-ice drift vectors with on-ice drifters trajectories (Sect. 3.3) for both the Northern and Southern Hemispheres (Sect. 3.2). Here we analyse the statistics from this validation.

Table 2 summarizes the validation results of all DM and S2S sea-ice drift vectors with a duration of 24 +/- 2 hours, thus the majority of DM vectors and a subset of all S2S vectors (right-panel on Fig. 4). The selected DM and S2S drift vectors thus correspond roughly to the same drift duration. In both Northern and Southern Hemispheres, S2S vectors result in better validation statistics than DM vectors. The reduction in the RMSE values ($\sigma$) is significant, e.g. a reduction by ~30% in NH and SH when adopting an S2S algorithm. The bias remains very small in NH, and is reduced in SH. As expected, the number of matchup samples $N$ is larger for S2S validation, and the increase with respect to DM is larger in NH than SH because most  on-ice buoys are from the central Arctic Ocean with many swath-to-swath overlaps (Fig. 2).

| | NH | | | | | SH | | | | |
|---|---|---|---|---|---|---|---|---|---|---|
| | $\mu_{dX}$ | $\mu_{dY}$ | $\sigma_{dX}$ | $\sigma_{dY}$ | N | $\mu_{dX}$ | $\mu_{dY}$ | $\sigma_{dX}$ | $\sigma_{dY}$ | N |
| **DM** | -0.09 | +0.01 | 1.36 | 1.32 | 2153 | -0.29 | +0.46 | 2.29 | 2.91 | 509 |
| **S2S** | -0.06 | -0.02 | 0.91 | 0.92 | 21245 | -0.04 | +0.20 | 1.36 | 1.62 | 3683 |

**Table 2:  Statistics from the validation of DM and S2S vectors derived from GCOM-W1 AMSR2 Ka-band imagery in Northern and Southern Hemisphere against on-ice buoy trajectories. Biases ($\mu$) and RMSE ($\sigma$) in dX and dY components of the drift vectors are reported in km, N is the number of matchup pairs. Only vectors with a duration of 24 +/- 2 h are validated.**

The validation statistics are somewhat worse in the SH than in the NH. This is probably due to the lower number of validation data points, and the fact that some are at the outskirts of the sea-ice cover, in dynamical regimes that are challenging for motion tracking from coarse resolution radiometry imagery (Fig. 1).

290 These numbers mean that a typical 24 h drift displacement vector can be measured with an uncertainty of typically 1 km in each component by an S2S product, even though the original imagery is at a rather coarse resolution (the -3dB footprint of the AMSR2 36.5 GHz channels is 7x12 km, see Table 1). As expected, the uncertainty of DM sea-ice drift vectors is larger (~30% increase). In the next section, we repeat the validation experiment but this time we consider the full range of durations, from ~100 min to 52 h.

### 4.3    Validation of sea-ice drift vectors with any duration

Fig. 5 documents the evolution of the validation statistics (bias and RMSE) with the drift duration for the same S2S and DM drift vectors based on GCOM-W1 AMSR2 Ka-band imagery. The bias (μ) and RMSE (σ) in dX and dY components of the S2S vectors are plotted for drift duration ranging from ~100 min (drift computed from two consecutive swaths) to 52 hours. Bias and RMSE of DM drift vectors are plotted for 24 h and 48 h drift duration separately. The values reported in Table 2 correspond to the conditions around the 24 h tick-mark.

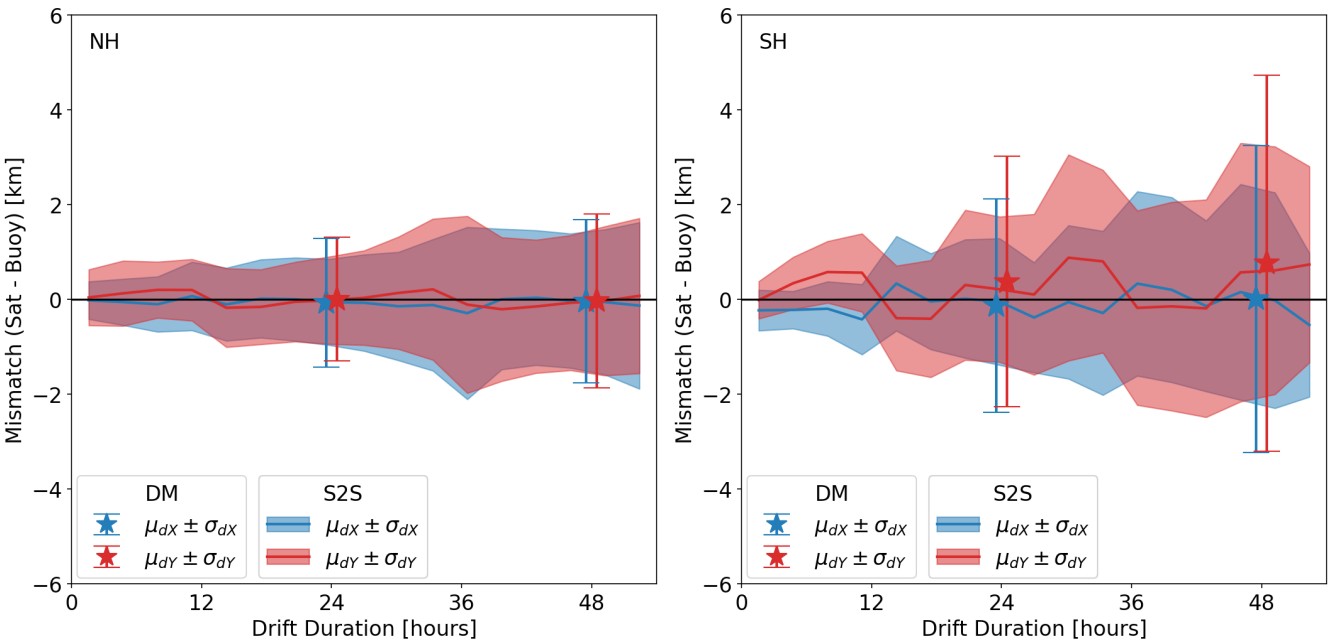

**Figure 5: Evolution of the validation statistics of the S2S and DM drift vectors with drift duration in NH (left) and SH (right). For S2S drift vectors, bias (solid lines) and RMSE (shaded regions) are plotted for the range from ~100 min to 52 h. Symbols (bias: star, RMSE: error bar) are shown for DM drift vectors with 24 h and 48 h drift duration.**

Fig. 5 confirms that the RMSE obtained for S2S drift vectors is smaller than that of 24 and 48 h DM vectors, and at intermediate drift durations below 52 h. The RMSE of S2S vectors increases regularly with the drift duration, from ~0.5 km for drift duration of ~100 min to ~2 km for drift duration of 52 h. In NH, the bias stays small for the whole range of drift

durations, and is smallest around the 12 h, 24 h, 36 h, and 48 h marks, a repeat cycle that we will study in more details in the next section. In SH, the same cycle is observed but with a stronger amplitude. As already reported in Table 2, bias and RMSE of both S2S and DM vectors are larger in the SH. In any case, we confirm that the S2S drift processing can be advantageous for passive microwave radiometry missions (with respect to the current state-of-the-art which are all based on DM processing), as S2S brings many more drift vectors (Sect. 4.1), and those drift vectors are more accurate (this section).

### 4.4    Seasonal evolution of the drift accuracy in the Arctic

The two last sections focused on two 3 months winter periods in the Arctic and Antarctic. Here, we present monthly validation results covering Oct 2019 to Dec 2020 (15 months) in the Arctic. Our main objective is to investigate if the S2S approach helps the retrieval of sea-ice drift vectors during the Arctic summer melt season. Due to surface melt and increased wetness in the atmosphere, the tracking of sea-ice drift from passive microwave instruments has traditionally been a challenge during summer.  While Kwok (2008) has shown that imagery from the AMSR2 mission can be used to track summer sea-ice drift (using a DM approach), the accuracy when compared to buoy trajectories was shown to be much reduced.

Fig. 6 shows monthly validation statistics for several DM and S2S products obtained from the AMSR2 36.5 GHz imagery. Both 48 h, 24 h and 18 h drift products were prepared and validated following Sect. 3.3. Fig. 6 confirms that the validation statistics of drift vectors with shorter durations (e.g. 18 h and 24 h) are better than those of vectors with longer durations (48 h), both in terms of RMSE and bias, and for the whole 15 months period. This was already noted in Sect. 4.3 for the period Oct-Dec 2019. Fig. 6 also confirms that, for most of the year, S2S drift vectors reach better validation statistics than DM vectors. This is true for all the winter months (Oct – Apr). However, the better accuracy of S2S drift vectors is not apparent during the summer months (May – Sept) when DM reaches (slightly) better results. Validation results during summer are indeed worse than during winter, but the main driver for the worsen accuracy in summer seems to be the duration of the drift vectors (24 h vs 48 h), not the adoption of an S2S vs a DM approach.

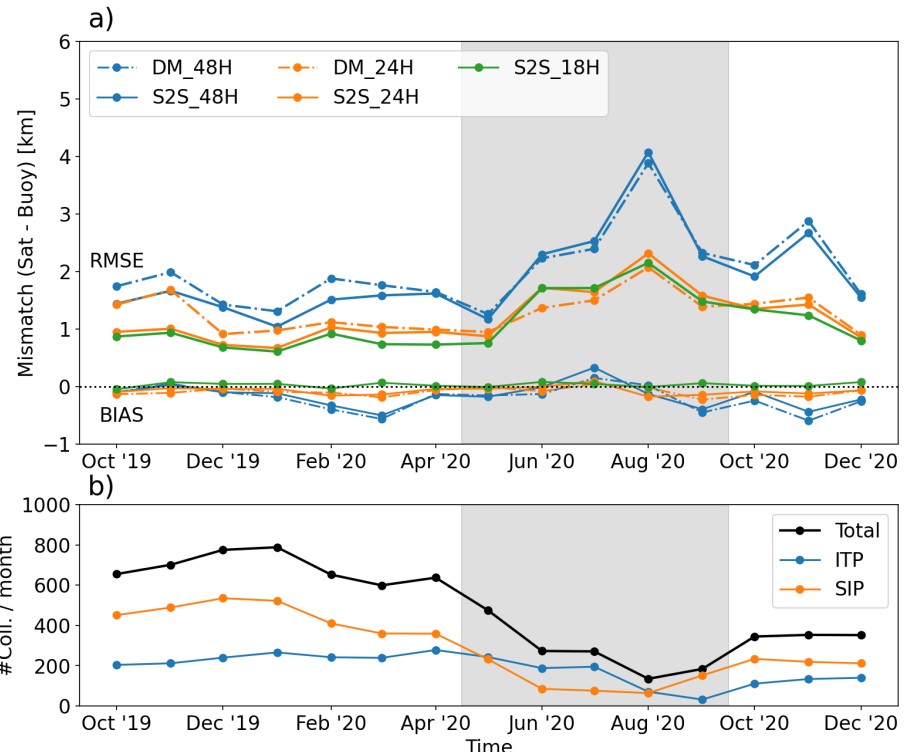

**Figure 6: a) Monthly validation statistics of the S2S and DM drift vectors with drift durations 48 h (blue), 24 h (orange) and 18 h**
**(S2S only) from Oct 2019 to Dec 2020 in NH. b) Number of collocation matchups per months for the DM products (black: total,**
**blue: Ice Tethered Profilers, and orange: seaiceportal.de). Both RMSE and BIAS are reported. The summer season (May-Sept) is**
**greyed.**

During summer in the Arctic, the atmosphere is wetter and contributes significantly to the brightness temperature recorded at

36.5 GHz, effectively hiding more of the surface emissivity. The surface emissivity is also more variable in time because of

the cycles of sub-daily cycles of melt/freezing (early and late in the summer season) and the direct impact of weather system

traveling over the sea-ice. It is thus not a surprise to see better validation statistics with shorter than longer drift durations

since a shorter duration will increase the chance of tracking the same surface emissivity patterns with less chances for a

change happening in the time between the two images.

When conducting the same investigations with the 18.7 GHz imagery of AMSR2 (not shown) we found roughly the same

results but the validation statistics were slightly worse than those obtained with 36.5 GHz throughout the year. The 18.7 GHz

microwave frequency is emitted from deeper in the sea ice and snow medium and is less affected by the atmosphere, so that

one would expect more stable surface emissivity patterns available for sea-ice motion tracking (Kwok, 2008). However the

coarser resolution of the 18.7 GHz frequency channels (Table 1) works against this property by blurring the emissivity

patterns.

We note that, even if DM vectors seem to validate better than S2S vectors during the summer melt season, adopting the S2S approach still gives many more vectors per day than the DM approach.

### 4.5     Impact of geo-location accuracy of the imagery

Geo-location accuracy of the satellite images from which drift vectors are computed is a key component of the uncertainty budget. Even very high resolution satellite images (such as those from SAR) will give poor sea-ice drift vectors if the images
are not correctly geo-located.

To simulate the impact of geo-location accuracy on the S2S drift vectors, we purposely mis-register the GCOM-W1 AMSR2 36.5 GHz imagery to the locations of the 18.7 GHz (Ku-band) imagery. Because of how the Ku- and Ka-band radiometer feeds are arranged in the focal plane of the AMSR2 instrument, they do not exactly point at the same locations on Earth during the rotation of the reflector antenna (Maeda et al. 2016). The Ku-band Field of Views (FoVs) are systematically
further ahead in the flight direction with respect to the Ka-band FoVs. The offset between the positions ranges from 750 m to 1 km in the flight direction, with a mean value of 917 m. For comparison, the spacing between successive scans of both microwave frequency channels is 10 km, and the spatial resolution of the Ku-band FoVs is 18 km (Table 1). The geo-location error introduced is thus of an order of magnitude less than both the spacing and resolution of the imagery channel.

We repeat the whole S2S and DM processing but this time remapping the Ka-band imagery mis-registered to the Ku-band
locations. A visual analysis of the new sea-ice drift maps does not identify obvious problems with the new vectors (not shown).

Fig. 7 is a repeat of Fig. 5 a) but with the Ka-band imagery using the geo-location of the the Ku-band FoVs. Comparing Fig. 7 to Fig. 5 a) informs on the impact of geo-location errors on the accuracy of S2S and DM drift vectors. The accuracy of DM vectors is only marginally impacted by the geo-location error. Their bias is still very close to zero, and their RMSE is
similar to those reported on Fig. 5 (around 1.5 km for 24 h drift duration and 1.9 km for 48 h). Conversely, the accuracy of S2S drift vectors is strongly impacted. Especially, the bias of the dX (*resp.* dY) component of the vectors varies between -1.5 km to +1.5 km (*resp.* -0.5 km to +0.5 km), with sharp transitions between positive and negative values around the 12 h and 36 h drift durations. Away from these transitions, i.e. for drift durations of 0 h, 24 h, and 48 h, the bias is much smaller and close to zero, as was the case in Fig. 5 a). The RMSE of S2S vectors (the width of the shaded areas) is similar to that
obtained on Fig. 5 a), across the full range of drift durations.

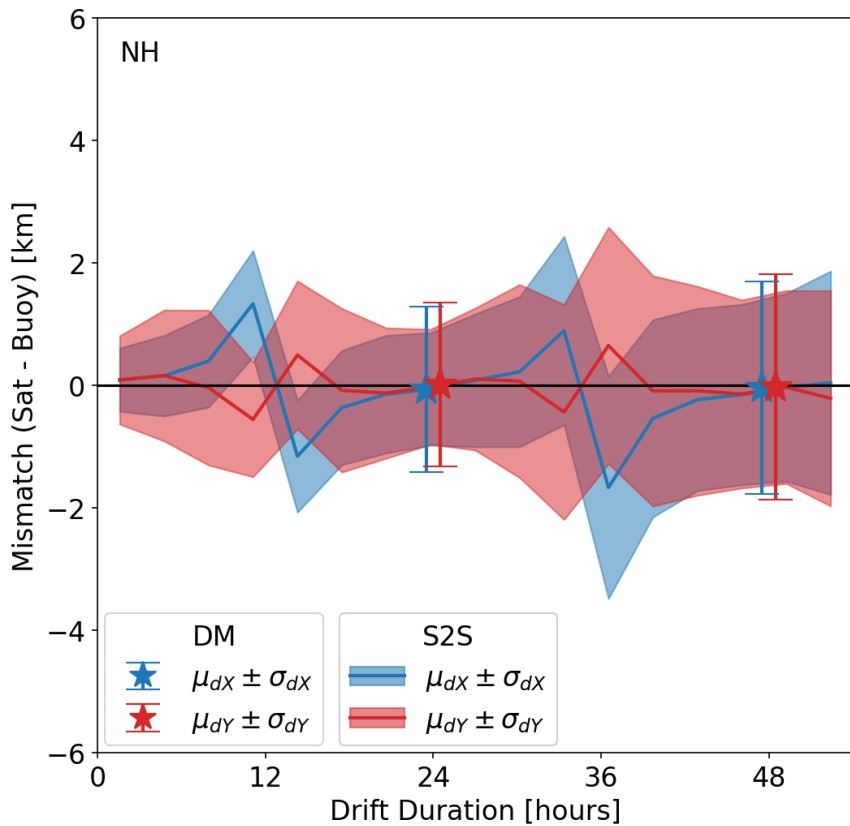

**Figure 7: Same as Fig. 5 a), but for the case where the 36.5 GHz imagery is purposely mis-registered to the location of the 18.7 GHz imagery.**

As documented in Fig. 7, the impact of geo-location error on the bias of the S2S drift vectors has a clear dependency on the duration of the drift vectors, with a repeat period of 24 h. This pattern is explained by the angle formed between the flight directions of the two orbits from which the S2S drift vectors are computed.

Geo-location accuracy is of importance for all satellite-based products, but will have an exacerbated impact on sea-ice drift products. Indeed, if both satellite images from which the drift vectors are computed have a geo-location error, and the two geo-location errors are in opposite directions, the drift vector will be strongly affected. Fig. 8 gives a schematic view of the impact of a constant geo-location error in the flight direction (which is very close to what we simulated in Fig. 7).

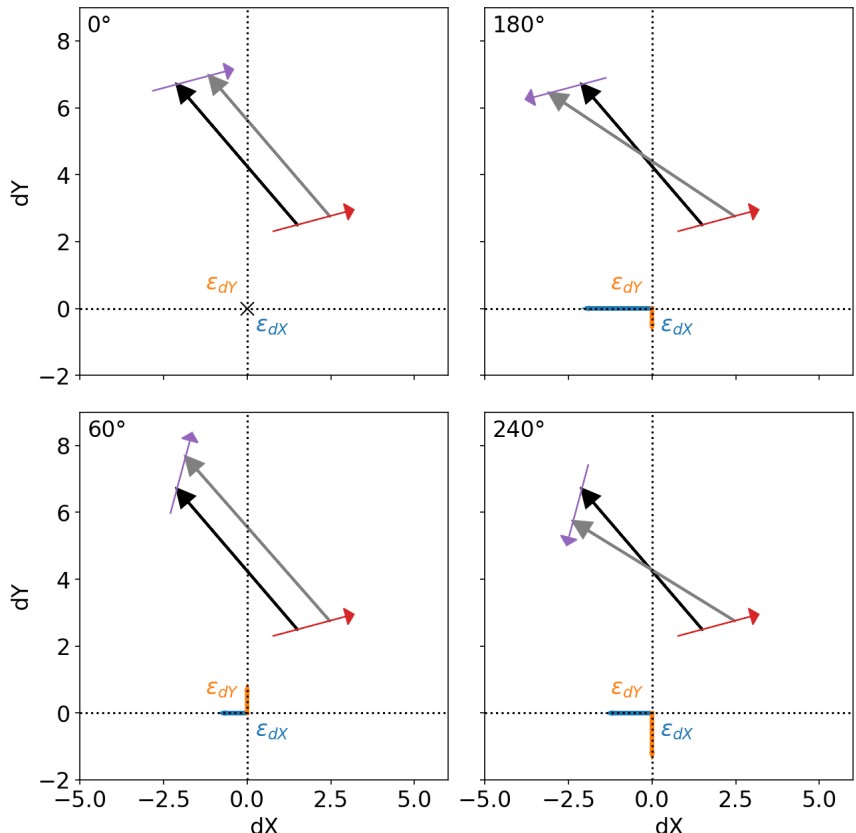

**Figure 8: Impact of a constant geo-location error in the flight direction on S2S retrievals. Each panel corresponds to a different relative angle θ_r between the two orbits from which sea-ice drift vectors are estimated. The true drift vector is in black, the geo-location offsets in the first and second orbits are of the same length, and create an erroneous drift vector (gray). The errors in the dX and dY components are ε_dX and ε_dY.**

In Fig. 8, we illustrate how the relative angle between two swaths sustaining the computation of an S2S vector impacts the magnitude of the retrieval error in the presence of a constant geo-location offset along the flight direction. The four panels correspond to four different relative angles. When the two swaths are close to parallel ($θ_r ≈ 0$), the geo-location error does not result in significant drift errors. This corresponds to time separation between the AMSR2 swaths of ~100 min, ~24 h, ~48 h. When the two swaths are in opposite flight directions ($θ_r ≈ 180$), the errors on the drift components are maximum, which is the case for time separation of ~12 h, and ~36 h. Other relative angles (two lower panels) have intermediate contributions to the errors. There is a direct link between the situations illustrated in Fig. 8 and the variation of the biases with drift durations in Fig. 7, in the case of a strong geo-location offsets in the flight direction. We already noted a similar cycle of the biases in the NH and SH validation exercises in Sect. 4.2 (Fig. 5) which might result from a small residual geo-

location error of the AMSR2 36.5 GHz imagery in the flight direction. In Sect. 5, we will discuss what the implications are for the retrieval of S2S drift vectors from the CIMR mission.

Before closing this section on geo-location errors, we discuss the fact that the DM drift vectors seem much less affected by the artificial geo-location error introduced for Fig. 7 than the S2S vectors are (compare the error-bar symbols of Fig. 5 and Fig 7, and especially that the bias is zero in both). When building daily composite maps of averaged $T_B$, the impact of the geo-location is smeared, but not canceled. Indeed, and especially at lower latitudes, locations on the sea-ice cover are often observed by sequences of subsequent orbits, which means a constant geo-location error from each orbit will still result in a

non-zero mean geo-location error in the daily averaged map. Nevertheless, the bias in drift components of DM vectors is close to zero because of the repeat cycle of the observations of (in our case AMSR2) orbits: a location on the sea-ice cover will be observed on average roughly at the same time of the day one or two days apart. This repeat cycle of the satellite orbit ensures that the non-zero mean geo-location error is similar at all locations in the daily maps, thus that the impact on the DM drift vectors is small ($\theta_r \approx 0$).

All in all, we conclude from this section that S2S drift vectors are more sensitive to geo-location errors than DM drift vectors are, and that the retrieval of accurate sea-ice drift vectors from individual swaths puts stringent requirements on the geo-location accuracy of passive microwave missions if all swath overlap pairs should be processed. Even seemingly limited geo-location errors such as those simulated in this section will result in inaccurate (biased) estimates when orbits overlap in opposite flight directions.

**4.6   Impact of the resolution and microwave frequency of the imagery**

The key principle of motion tracking algorithms (including those for sea-ice drift) is to identify local intensity patterns on one image, and track it on another image. The accuracy of the motion vectors will thus be better if the satellite images offer intensity patterns that are clearly defined and stable with time. The sharpness and stability of satellite microwave images of the sea-ice surface depend both on the spatial resolution achieved by the FoV, and the microwave frequency and polarization

of the imagery channel being processed. Indeed, channels (or instruments) with coarser spatial resolution will provide blurred images of the microwave emissions, with less or weaker intensity patterns to track. Channels with higher sensitivity to stable characteristics of the surface (e.g. sea-ice type, snow-depth, etc…) and/or lower sensitivity to the varying atmosphere above the sea ice (e.g. cloud liquid water path, air temperature) will offer sharper and more stable intensity patterns to track.

We investigate the impact of resolution and microwave frequency of the imagery by running an experiment where GCOM-W1 AMSR2 Ka (36.5 GHz) and W (89.0 GHz) imagery channels are first resampled on a pan-Arctic grid with spacing of

3.125 km. Such a fine spacing is an oversampling of the true resolution of both microwave channels (see Table 1). The remapping of each channel is done independently and accounts for the width of the FoV. Swaths are resampled separately, and S2S drift vectors are computed, collocated against buoy data, and validation statistics (bias and RMSE) extracted for the

same 3 month period as in Sect. 4.1. Only the ~24 h S2S vectors are computed. This experiment is then repeated for purposely coarsened versions of the images. Bi-linear coarsening kernels at factors 2 (6.25 km), 3 (9.375 km), 4 (12.5 km), 5 (15.625 km), and 6 (18.75 km) are applied to the original 3.125 km grid spacing images. S2S drift vectors with durations ~24 h are computed, collocated to buoys, and their validation statistics extracted.

Fig. 9 plots the evolution of the validation statistics (bias and RMSE) of the AMSR2 NH ~24 h S2S drift vectors with the

width of the coarsening kernel. The best image resolution is to the right of the x-axis.

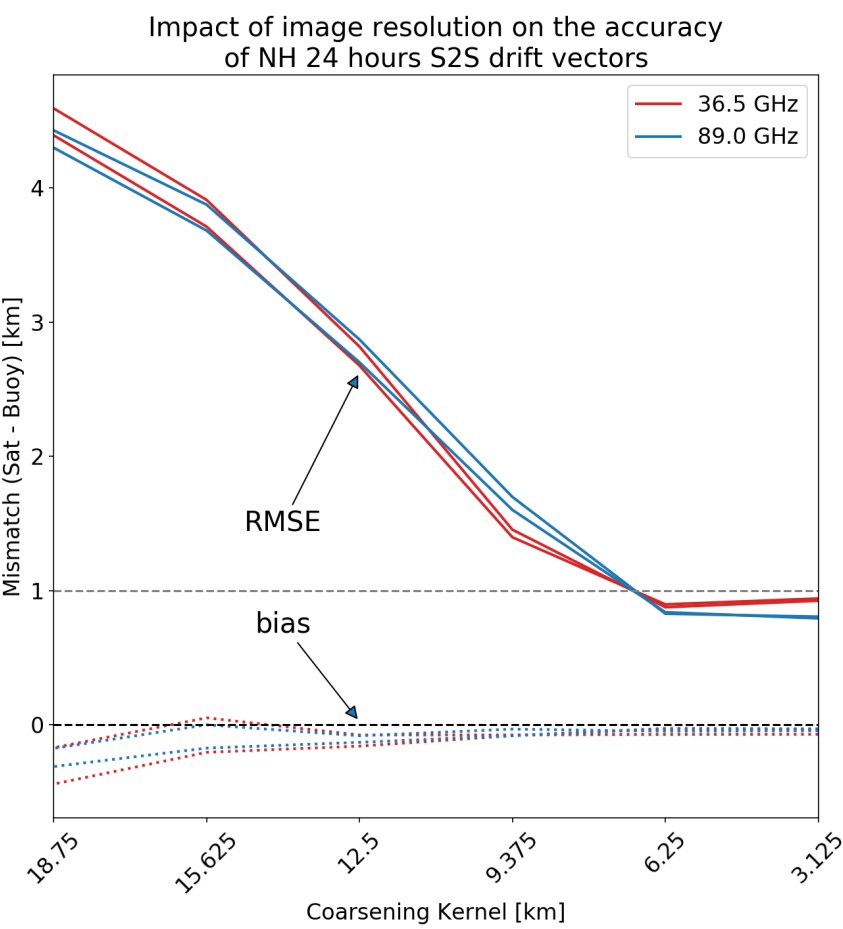

**Figure 9: Bias (dashed lines) and RMSE (solid lines) of NH ~24 hours S2S drift vectors processed from GCOM-W1 AMSR2 36.5 GHz (red) and 89.0 GHz (blue) as a function of the width of an image coarsening kernel applied on 3.125 km images. The best image resolution is to the right of the x-axis. The lines do not distinguish between the dX and dY components of the drift vectors as they evolve in a similar manner.**

As expected, the RMSE from both 36.5 GHz and 89 GHz imagery drops with finer spatial resolution (from left to right in the plot) and the bias is reduced towards zero. This behavior stalls at the 6.25 km kernel, and no further improvement of RMSE is observed when processing the full resolution 3.125 km images. In fact a slight worsening can be observed for the 36.5 GHz channels.

Thanks to their higher microwave frequency, the GCOM-W1 AMSR2 89 GHz imagery channels have a distinctly better

resolution than the 36.5 GHz channels (Table 1). However, this finer resolution does not seem to translate into a much better accuracy of the S2S drift vectors, as the RMSEs and biases reported on Fig. 9 are very similar for both microwave frequencies. Here we can hypothesize that the finer spatial resolution of the 89 GHz channels are balanced by the relative lack of intensity (and/or temporal stability) of intensity patterns to track over a ~24 hours period. Notably, the 36.5 GHz imagery is emitted from deeper in the ice/snow media, is more sensitive to sea-ice type and snow-depth (rather stable in

time), and less sensitive to the atmosphere above the ice. The 89 GHz is emitted closer to the top of the ice/snow media and is thus more sensitive to temperature changes. It is also much more sensitive to liquid water path in the atmosphere, that can easily blur or hide intensity patterns at the surface. We note that the lack of improvement of the RMSE metric from 6.25 km to 3.125 km could also be due to our rather simple approach to resampling the 89 GHz channels to the 3.125 km grid. However, more advanced gridding techniques (e.g. Backus-Gilbert) could also be challenged by the lack of sufficient

overlap between neighboring 89 GHz Field of Views. The slight increase of RMSE seen for the 36.5 GHz channels from 6.125 km to 3.125 km also indicates that overly oversampling the native spatial resolution of the imagery channels is not beneficial, and that it seems more efficient for sea-ice drift retrievals to aim at an optimum resampling resolution for each channel, rather than the finest possible spacing.

Still, the spatial resolution of the images does not only drive the accuracy of the drift vectors, but also the overall resolution

of the vector field. Indeed, a finer image will allow motion tracking algorithms to be run with smaller image sub-windows, and thus allow the creation of a denser vector field. For example, the finer resolution of the AMSR2 89 GHz channels allows for tracking smaller areas down to 70 km x 70 km, and processing denser vector fields (e.g. 31.25 km, Ezraty et al. 2007). Increasing the grid spacing to 31.25 km was attempted in our study but did not result in significantly different results for either the 36.5 GHz or 89 GHz imagery (not shown). In fact, the result was a slightly larger RMSE for both frequency

channels, and at all coarsening kernels. This might be because tracking smaller image sub-windows leads to tracking fewer intensity patterns per vector. As is true for the retrieval of many other geophysical variables, increasing the spatial resolution often leads to increases in the noise level. The increased RMSEs might also reflect that most of our validation data are from the central Arctic Ocean and mostly sample drift events that are coherent over large spatial scales. The higher resolution drift vectors might be more accurate in cases where sharp spatial gradients of the drift field are observed, either in the event of

sea-ice deformation, or in places with stable velocity gradients such as in the East Greenland Sea. However these conditions are not well enough represented in our validation data set to have a positive impact on the overall statistics.

## 4.7 Simulating the spatial coverage of S2S drift vectors from CIMR

The wider swath of CIMR with respect to AMSR2 (and to all other conically-scanning radiometer missions) should result in larger areas of overlap between individual swaths, and thus in more S2S sea-ice drift vectors. A larger number of S2S vectors should be observed towards lower latitudes (needed to better sample the core of the sea-ice cover in the Southern Hemisphere) and higher latitudes (needed to monitor the sea-ice cover close to the North Pole).

We simulate 48 hours of CIMR orbit and swath coverage (Sect. 3.4) and enter these swaths in our S2S sea-ice drift processing chain. We repeat the analysis of spatial distribution presented in Sect. 4.1 with AMSR2 data.

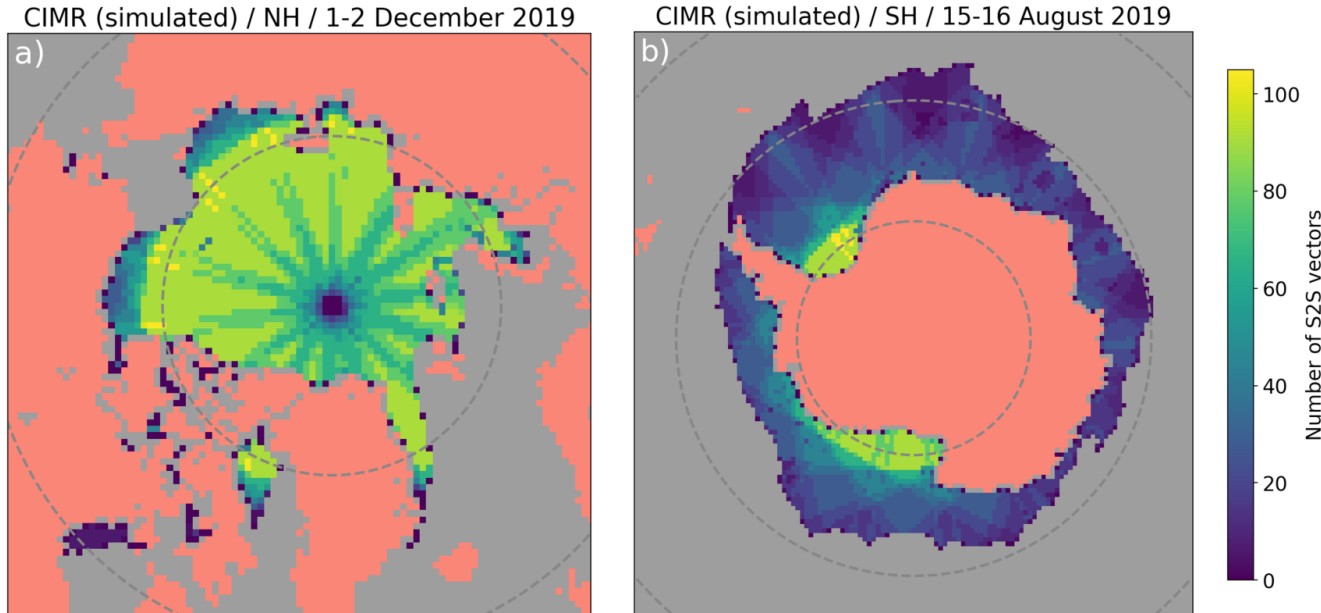

**Figure 10: Same as Fig. 3 but for simulated CIMR coverage. a) number of S2S vectors per grid cell in the Northern Hemisphere with the same sea-ice cover as 1-2 December 2019. b) number of S2S vectors per grid cell in the Southern Hemisphere and with the same sea-ice cover as 15-16 August 2019. Parallels at +/- 75 and +/-60 are drawn.**

Fig. 10 plots the spatial distribution of the number of S2S vectors obtained from the simulated CIMR orbit data. To ease comparison with AMSR2, we use the same sea-ice cover as in Fig. 3 (1-2 December 2019 in NH, 15-16 August 2019 in SH). The total number of S2S vectors for the Arctic coverage is 143,101 for CIMR (compared to 110,400 for AMSR2 and 1,729 for the DM product). Over Antarctic sea-ice, the number is 108,601 for CIMR (compared to 74,737 for AMSR2 and 3,617 for the DM product). The wider swath of CIMR thus results in 30-40% more S2S vectors compared to AMSR2.

When comparing Fig. 10 (CIMR) to Fig. 3 (AMSR2) in the Northern Hemisphere, a striking improvement is seen towards the North Pole. In Sect. 4.1 we explained how the polar observation hole of the AMSR2 imagery, despite being of only 0.5° latitude, has a wider impact on the coverage of S2S vectors near the pole. The CIMR orbit and swath width are specifically

optimized to allow full sub-daily coverage of the polar regions including the poles, thus "no hole at the pole" (Donlon et al., 2020). Our analysis shows that the number of S2S vectors from CIMR reduces towards the pole, but that there are vectors all the way to the pole on a daily basis.

We underline that the actual number of S2S vectors from CIMR will be larger than discussed here, simply because the increased spatial resolution of the Ku- and Ka-band imagery (Table 1) will allow a denser vector field (target 25 km grid-

spacing, Donlon et al. 2020) than we processed (62.5 km grid-spacing).

## 5  Discussion

In this study, we processed Arctic and Antarctic sea-ice drift vectors from the AMSR2 imagery, and compare them to GPS trajectories of on-ice buoys. We use a state-of-the-art algorithm (Lavergne et al., 2010), as implemented in the processing chain of the EUMETSAT OSI SAF. Our focus is to compare two approaches: a) processing sea-ice drift vectors from the

intersection of individual swaths (S2S-type products) and b) the current status-quo which is to process sea-ice drift vectors from daily average maps of satellite signal (DM-type products). We show that not only many more vectors can be processed with an S2S-type product on a daily basis (Sect. 4.1), but that these vectors also validate better against GPS trajectories of on-ice buoys (Sect. 4.2 and 4.3) during winter. This is not systematically the case during summer season when it is more important to target shorter drift durations (e.g. 18 h and 24 h vs 48 h) than to adopt an S2S approach (Sect. 4.4). We also

document the impact that geo-location uncertainty (Sect. 4.5) and spatial resolution of the imagery (Sect. 4.6) have on the accuracy of the sea-ice drift field.

We link the better accuracy of the S2S vectors to two aspects of the motion tracking algorithm. First, the imagery of individually-gridded swaths presents sharper intensity features over sea ice, which results in tracking more accurate sea-ice motion vectors. Once they are averaged over a daily period, the satellite imagery is blurred by the very motion we want to

measure. Second, the start and end times assigned to S2S vectors are more accurate than those assigned to DM vectors. Better defined start and end times lead to better collocation with buoy trajectories, which improves the validation statistics. Beyond validation, having more accurate start and end times will help matching S2S vectors with (e.g. hourly) fields from ocean/ice forecast models. This will have most impact in cases where the motion field has strong temporal gradients, e.g. when a low-pressure system travels over sea ice.

To the best of our knowledge, the better accuracy of S2S vectors from passive microwave satellite missions has not been systematically documented. Maslanik et al. (1998) note that investigations on S2S vectors had been conducted but that this *does not significantly improve the comparison with buoys*. It must be underlined that we now have acccess to much better data than in the late 1990s, both in terms of satellite imagery (AMSR2 versus the SSM/I) and in terms of buoy data. Indeed,

most Arctic buoy data in the late 1990s were 3-hourly trajectories with geo-location via the Argos positioning system, which
could have degraded the validation statistics. The AMSR2 mission also achieves a better spatial resolution than the SSM/I.

Existing routinely generated sea-ice drift products such as those from the EUMETSAT OSI SAF, IFREMER CERSAT, or JAXA's AMSR2 ground processor could readily move from DM sea-ice drift products to an S2S configuration with limited investment, since the core of motion tracking algorithm is the same. The positive impact will include better accuracy, better timeliness, and more sea-ice drift information for their users. Both EUMETSAT OSI SAF and IFREMER CERSAT products
have time duration ranging from 2 to 3 days for their DM products, while users would favour drift vectors with durations closer to 24 hours. Our results (e.g. Fig. 5) show that S2S products from AMSR2 can be moved towards 24 hours and shorter which can be easier for Data Assimilation applications.

When it comes to new satellite missions and services, and specifically CIMR, we recommend that S2S sea-ice drift products are adopted from the start in the ground segment, so that the processing chains are dimensioned accordingly and users and
downstream services can prepare for it.

The CIMR mission has several characteristics that can be exploited to prepare sea-ice drift information with a higher accuracy and resolution than currently available from any passive microwave satellite mission. We discuss some of these here. First, CIMR will provide a suite of microwave frequency channels at a much higher spatial resolution than AMSR2 (see Table 1), e.g. "better than 5 km" at 36.5 GHz (Ka-band), and 5 km at 18.7 GHz (Ku-band). In Sect. 4.6, we illustrated
that imaging resolution is a key element for the final accuracy of sea-ice drift vectors (Fig. 9) and for the spatial resolution of sea-ice drift products (the spacing between neighboring vectors). Particularly, we showed that the high-frequency channels of the AMSR2 instrument (89 GHz) did not bring extra accuracy, probably because the radiation was emitted at a shallower depth in the snow/ice layer, and the resulting intensity patterns were less stable with time and thus harder to cross-correlate. As for the retrieval of other ocean and sea-ice parameters, the true advantage of CIMR will not be to access to high-
resolution radiometric images in absolute terms, but rather to give access to high-resolution radiometric images at microwave frequencies that until now were only accessible at medium-to-coarse resolution (> 10 km). While the 36.5 GHz channels will make the main contribution to the sea-ice drift accuracy throughout the year, the 18.7 GHz channels at 5 km can contribute during the summer melt season (Kwok, 2008). In any case, in order to fully succeed as a sea-ice drift mission, CIMR will also have to deliver a high geo-location accuracy. In Sect. 4.5, we documented how geo-location errors,
especially if they are systematic with respect to the imaging dimensions, rapidly grow into prohibitive retrieval errors for S2S-type of products. In this respect, S2S products are more affected than DM products, because the latter average the geo-location error throughout the day and -due to the orbital configuration of most polar orbiting missions- the same areas are revisited roughly at the same times a few days apart. Since CIMR will use a large rotating deployable mesh antenna reflector, the geo-location accuracy translates into a stringent requirement on the pointing accuracy of the antenna. In

addition, it is expected that small remaining systematic geo-location errors will be asessed and corrected against coastlines (Wiebe et al., 2008). From the results presented here, we argue that the accuracy (especially the bias) of a future S2S sea-ice drift product from CIMR, assessed against on-ice GPS buoy trajectories, would constitute an independent check of the geo-location accuracy of the mission, e.g. as part of the Calibration Validation (Cal/Val) phase.

The high spatial resolution, and high geo-location accuracy CIMR Level-1B (geo-located brightness temperatures in swath projections) products will not directly be input to the Level-2 S2S drift processing. Indeed, the motion tracking algorithms require remapped brightness temperatures as input. Following other missions (e.g. NASA SMAP), CIMR will also process a Level-1C product: individual swaths of brightness temperatures remapped to a fixed Earth-referenced grid. It is foreseen that CIMR's Level-1C product will be the input of its Level-2 S2S drift product, which has several advantages including the collocation of all CIMR's channels onto a set of directly overlapping EASE2 grids (Brodzick). In Fig. 8, we found that the accuracy of the drift vectors stalled (89 GHz) and even slightly degraded (36.5 GHz) when grid spacing approached or went beyond the true resolution of the imagery, possibly because our gridding methodology was basic and might have introduced artefacts at small grid spacing. The gridding algorithms implemented in the CIMR Level-1C product should be carefully designed to not introduce such artefacts and retain the true resolution of the Level-1B information, so as not to reduce the final accuracy of the Level-2 S2S drift product.

A number of features of the CIMR mission prompt specific research and development before they can be fully exploited in the operational sea-ice drift processing. We mention some of these here, as a way forward for the development of CIMR-specific algorithms. CIMR will offer high-resolution brightness temperature imagery at the microwave frequencies listed in Table 1. Being the first radiometer to offer such a high spatial resolution at, e.g. 6.9 GHz and 10.7 GHz, the potential contribution of these frequency channels to a sea-ice drift product will have to be investigated. Since they are emitted from deeper in the snow and sea-ice layer, they could potentially contribute to the summer melt period, in addition to 18.7 GHz.

CIMR will offer both a forward and a backward scan, separated with four minutes at the center of the swath (Donlon et al., 2020). Considering the typical sea-ice drift speed and the resolution of the CIMR channels, we do not expect to be able to detect motion taking place during such a short time. However, the swaths corresponding to the forward and backward scans are independent, overlapping, and mostly simulatenous images that can both be input to a sea-ice motion tracking algorithm. This might be used to reduce the noise of retrieved sea-ice drift vectors, either by averaging the four independent drift vectors (forward-forward, forward-backward, backward-forward, and backward-backward pairs), or as an additional input to the quality control step and the detection of rogue-vectors (see Sect. 3.1).

Independently of the R&D elements outlined above, the design and dimensioning of a CIMR Level-2 sea-ice drift Payload Data Ground Segment (PDGS) will need careful consideration for the specificities of a S2S sea-ice drift product. Typically, a
Level-2 processing chain produces a single Level-2 product per input Level-1 file. This will not be the case for the CIMR Level-2 drift product. Instead, each incoming Level-1C file can potentially be paired with several past Level-1C files, and each pair results in a Level-2 sea-ice drift product. When dimensioning the data throughput of the sea-ice drift processor, we must make sure that all relevant pairs are processed before the next Level-1C file is available. One can think of several schemes for selecting those interesting pairs, including the drift duration (time separation between the two Level-1C files)
and the angle between the swaths, which may relate to geo-location noise (see Sect. 4.3). The setup we adopted in this study (pairing a Level-1 file with all preceding Level-1 files within 48 hours) resulted in about 40 Level-2 sea-ice drift files per Level-1 file (>500 Level-2 files per day) which can be overwhelming, but does fully sample the temporal variability of the sea-ice motion field. In preparing for the CIMR mission, one also has to consider that today's users, especially from the modelling community, are not used to these S2S products. Adopting this type of product might require some dedicated
efforts, that can luckily be conducted in advance, for example using AMSR2 data as we did here. As with the other parameters to be observed by CIMR, a Level-3 sea-ice drift product should be prepared that optimally combines e.g. a day's worth of S2S products (having very different time durations) into a complete map of e.g. ~24 hours sea-ice motion vectors. To the best of our knowledge, such merging algorithms do not exist. Finally, we note that sea-ice drift vectors are today processed at high resolution from Sentinel-1 SAR images, e.g. in the Copernicus Marine Environment Monitoring Service
(CMEMS), and that this type of SAR-based product will continue with other missions such as the Radarsat Constellation Mission (RCM), the Expansion Sentinel (formerly known as High Priority Candidate Mission) Radar Observing System for Europe – L-Band (ROSE-L), and later the Sentinel-1 "next generation" platforms. Despite the increase in SAR missions, the coverage of the Arctic and Antarctic sea-ice at a sub-daily frequency will remain prohibitive in the foreseeable future. A Level-4 Ice Drift Analysis (IDA) product merging S2S drift products from both CIMR and several SAR missions would fill
a data gap by providing a high-level sea-ice motion product at high resolution and accuracy, with a daily complete coverage. Such a Level-4 product does not exist today. It could be developed from AMSR2 and Sentinel-1 data in preparation for the Copernicus Expansion Sentinels polar missions.

## 6    Conclusions

We investigate the feasibility and impact of adopting a "swath-to-swath" (S2S) versus "daily-map" (DM) framework for the
processing of sea-ice motion from modern passive microwave satellite missions such as JAXA's AMSR2, in preparation for the future CIMR mission. We find that S2S sea-ice drift vectors obtained from AMSR2 imagery are more accurate than the corresponding DM vectors when compared to GPS trajectories from on-ice buoys, both in the Arctic and Antarctic. An S2S configuration also results in many more drift vectors on a daily basis: the number varies with latitude and depends on the orbital and swath characteristics of the satellite mission. Since S2S drift vectors can be prepared for each new incoming

swath, this configuration yields much better timeliness, which is beneficial for several operational applications such as support to navigation and short term sea-ice forecasting. One potential limitation to the S2S configuration is that it is more sensitive to inaccurate geo-location, especially if the geolocation errors are systematic (e.g. a shift in the flight direction).

As far the the CIMR mission is concerned, we recommend the adoption of a S2S configuration for the Level-2 sea-ice drift product in the operational ground segment. Considering the microwave frequency channels, target spatial resolution, swath

width, and geolocation accuracy specified for the CIMR imagery, its Level-2 sea-ice drift product will allow unprecedented spatial resolution, coverage and accuracy for a microwave radiometer mission. Several other new characteristics of the CIMR mission (e.g. the relatively high spatial resolution at 6.9 and 10.8 GHz, the backward and forward scans) will also contribute to an enhanced sea-ice drift product, but this requires further research.

We finally note that such Level-2 S2S sea-ice drift products will be new to a large fraction of the user community, and their

downstream uptake must be prepared. This includes the preparation of Level-3 daily products from the CIMR mission only, as well as Level-4 daily products merging several sources such as CIMR, SARs (e.g. Sentinel-1 and ROSE-L) and potentially on-ice buoy trajectories. Such Level-4 Ice Drift Analysis (IDA) will require the development of dedicated algorithms and processing chains.

**Data availability**

A subset of the S2S and DM data used in this study were made available for inspection during the review process. The data are hosted at the Norwegian Meteorological Institute. They are formatted as NetCDF4 (classic) files, and follow the Climate and Forecast (CF) and Attribute Convention for Data Discovery (ACDD) conventions. All sea ice-drift products made available here are from the GCOM-W1 AMSR2 36.5 GHz imagery. The following data were prepared:

Northern Hemisphere covering 15 to 30 November 2019:

• S2S drift vectors (https://doi.org/10.21343/q1e3-1489);

• DM drift vectors (https://doi.org/10.21343/dts5-bf20);

Southern Hemisphere (Weddell Sea) covering 15 to 31 July 2019:

• S2S drift vectors (https://doi.org/10.21343/0asd-6t60);

- DM drift vectors (https://doi.org/10.21343/yfj4-2528);

We selected those two periods because they exhibited dynamic events in the sea-ice drift fields, including sharp spatial gradients and rotation patterns caused by low pressure systems. This should help demonstrate the different characteristics of the DM and S2S approach.

In addition, we will make available the 24 h Northern Hemisphere S2S and DM drift products covering the period October 2019 to December 2020 (15 months) at the same data center.

**Competing Interests**

The authors declare that they have no conflict of interest.

**Author contribution**

TL designed and carried out the study. MPS performed the CIMR orbit simulations. ED optimized and operated some of the sea-ice drift software. TL prepared the manuscript with contributions from all co-authors.

**Acknowledgements**

The AMSR2 Level-1 data were accessed through JAXA's Global Portal System (https://gportal.jaxa.jp/gpr/, last accessed 26th february 2021).

The Ice-Tethered Profiler data were collected and made available by the Ice-Tethered Profiler Program (Toole et al., 2010; Krishfield et al., 2008) based at the Woods Hole Oceanographic Institution (http://www.whoi.edu/itp, last accessed 1$^{st}$ June 645 2020). A variety of other buoys were accessed from the data portal meereisportal.de (Grosfeld et al., 2016, last accessed 1$^{st}$ June 2020), including all Antarctic buoys, and the buoys deployed at and around the Multidisciplinary drifting Observatory for the Study of Arctic Climate (MOSAiC) site.

This study was suported by ESA through the CIMR Mission Requirement Consolidation study, and the Climate Change Initiative Sea Ice Phase 2 project.


We used a sea-ice drift processing chain developed at MET Norway as part of the EUMETSAT OSI SAF service.

Our gratitude goes to the open-source community at large, and especially the maintainers of the Python language and its modules (numpy, scipy, matplotlib, pytroll,…).

This project took advantage of NetCDF software developed by UCAR/Unidata (http://doi.org/10.5065/D6H70CW6, last accessed 23rd October 2020).

Trygve Aspenes and Atle Sørensen, at the Norwegian Meteorological Institute, helped with accessing and pre-processing AMSR2 swath data.

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
