# Peer review of "Towards a swath-to-swath sea-ice drift product for the Copernicus Imaging Microwave Radiometer mission."

_The Cryosphere, 2020_

## Referee Comment (RC1) · Anonymous Referee #1 · 10 Dec 2020

Conventionally, sea ice motion from passive microwave observations is extracted from aggregated brightness temperature daily products covering the entire Arctic or Antarctic domains.

This paper investigates the possibility of deriving sea ice motion vectors directly from the overlapping AMSR2 individual swaths (S2S scenario) as opposed to the daily products (DM scenario) and implications on the future ESA CIMR mission. A well-established ice motion tracking algorithm based on the Continuous Maximum Cross-Correlation (CMCC) approach was applied to derive ice motion vectors in both the S2S and DM scenarios. The authors demonstrated that a much larger number of ice motion vectors with higher accuracy (as validated against in-situ buoys) is derived in the case of S2S compared to DM scenario. The S2S ice motion extraction scenario is recommended to be applied to the future CIMR mission, which will provide a higher spatial resolution compared to AMSR2. This is an interesting paper, but I have the following comments which need to be addressed before the manuscript can be considered for publication.

Major comments:

1.  In this study, the authors used only winter time periods for both the Arctic and Antarctic. What about the summer time? Could S2S approach provide better (or any reasonable) ice motion tracking results compared to the DM approach in summer time? Would lower remote sensing frequencies be recommended in that case (due to the larger penetration depth) as opposed to the higher frequencies? I think the paper will look much better if quantitative evaluation of ice motion provided by S2S versus DM during the summer time is presented.

2.  The authors discuss the differences in sea ice motion tracking from different frequency channels (mainly Ka and W due to their relatively high spatial resolution). However, polarization options were not discussed. What are the differences in terms of the number and accuracy of ice motion vectors derived from the horizontal and vertical polarization swaths? What optimum polarization option or polarization combinations are recommended for the ice motion tracking?

Technical corrections:

There is some language inaccuracies in the paper. I tried to point out some of them below with suggested changes.

Consider to mark figure panels with letters (a), (b), etc.

Line 37. "These can…". It seems that some word between "These" and "can" is missing.

Line 194. "over a Northern and a Southern Hemisphere grid.". Should "a" be replaced with "the"?

Line 232. "A first" → "The first".

Line 234. "…very different characteristics to the DM products…" → "…very different characteristics **compared** to the DM products…"

Line 242. "…these mean times associated with the DM ice drift product are averaged values…" → "…these mean times associated with **the fact that** the DM ice drift product are averaged values…"

Line 278. "…the low number of validation data…" → "…the **lower** number of validation data **points**…"

Line 284. "…but this time studying…" → "…but this time **we consider**…"

Line 285 and 289, and throughout the text. "100 mn → "100 **min**".

Line 318. "Fig. 6 is a repeat of Fig. 5… ". In fact, Fig. 6 is similar to Fig. 5 (left, NH) and not the entire Fig. 5. Please reflect it accordingly in the text.

Fig.7 and Fig.8. Please move the figure title to the figure caption.

---

## Author Comment (AC1) · 19 Dec 2020

**Data availability**

A subset of the S2S and DM data used in this study are made available for inspection during the review process. The data are hosted at the Norwegian Meteorological Institute. They are formatted as NetCDF4 (classic) files, and follow the Climate and Forecast (CF) and Attribute Convention for Data Discovery (ACDD) conventions. All sea ice-drift products made available here are from the GCOM-W1 AMSR2 36.5 GHz imagery. The following data are prepared:

Northern Hemisphere covering 15 to 30 November 2019:

- S2S drift vectors (https://doi.org/10.21343/q1e3-1489);

- DM drift vectors (https://doi.org/10.21343/dts5-bf20);

Southern Hemisphere (Weddell Sea) covering 15 to 31 July 2019:

- S2S drift vectors (https://doi.org/10.21343/0asd-6t60);

- DM drift vectors (https://doi.org/10.21343/yfj4-2528);

We selected those two periods because they exhibited dynamic events in the sea-ice drift fields, including sharp spatial gradients and rotation patterns caused by low pressure systems. This should help demonstrate the different characteristics of the DM and S2S approach.

---

## Referee Comment (RC2) · Anonymous Referee #2 · 21 Dec 2020

Summary

This paper presents an approach to calculate swath-to-swath (S2S) sea ice motion vectors from passive microwave imagery. Via comparisons with buoys, this approach is shown to be more accurate that the standard daily map products that composite brightness temperatures over a 24-hour period. The S2S are improved because the TB values are instantaneous instead of a "blurred" average and the time between images is exact as opposed an average time of all passes, which also "blurs" the motion estimates. The methodology is promising for the future CIMR mission, which will have wider swaths to obtain more motion vectors and higher spatial resolution for greater

accuracy.

General Comment

Swath-to-swath sea ice motion vectors has been a long-discussed idea, so it is great to see it successfully implemented here. The methodology is sound and it appears to be viable to do operationally. The paper is well-written and the results are convincing. I have a couple general comments. First, there is discussion of the packaging of the fields in Section 5, but it's still not totally clear how this would happen. I can understand the baseline approach, where there are fields for each of the overlaps and the times. But also discussed is the generation of daily maps. I agree that this would be useful and would probably be most convenient for most users. But it's not clear how this would be created. You would have motions for time periods of ∼100 minutes to over 24 hours. How would the different time separations be combined? The ice motion will vary between the periods, so simple interpolation/extrapolation may not work. The easiest thing would be to use the repeat orbit overlaps from each day – then there would be 24 hours between all vectors. But, of course, this leaves out many vectors. Maybe there could be some kind of weighting scheme to optimally combine vectors over different time intervals into an optimum cohesive daily map.

Another thing that came to mind while reading is the potential utility for summer motions. It's understandable to focus first on winter, but summer is not mentioned until Section 5. There are well-known limitations to using PM TBs for summer – most notably the surface melt and (especially for 89 GHz) greater water vapor levels. The S2S approach seems like it would be potentially quite helpful. First, the exact time of S2S will remove some error because ice is more dynamic in the summer, and potentially improve accuracy of the more sensitive lower frequency channels. For example, 18.7 GHz S2S may obtain better summer motions than daily composites. On the other hand, I wonder if the instantaneous S2S fields might cause some problems for the 89 GHz channels because water vapor can change rapidly and the "smearing" of the daily composite TBs may filter out some of that variability that could cause errors. It would
be great to have a summer example in the paper, but I think keeping the focus tighter on the winter case studies makes sense here. But I think some brief discussion of the limitations of PM for summer motions, e.g., in the Introduction, and a little more discussion in Section 5, would be helpful.

A few other minor comments are noted below. These are addressable in my view with minor revisions.

Specific Comments (by line number):

43: "short-lived" is ambiguous here. It may suggest something that lasts only a few weeks, but buoys can last at least a few years. That's short compared to long-term climate monitoring, but longer than what I would call "short-lived".

44: "scattered with vast distances between them" can be described more simply as "sparse". Perhaps rephrase this whole sentence to something more like: "Buoys have a limited lifespan before they exit out of the Arctic or the ice melts; this and limited opportunities for deployment result in sparse spatial coverage of the Arctic."

76-77: I don't see a reason to abbreviate "Section" here – it's more readable without the abbreviation.

81: Is there a specific citation recommended? At the least the website should be given, but if at all possible a formal citation (with date of access for a website) should be used.

Table 1: I would recommend giving the diameters of both dimensions of the sensor footprints, e.g., A x B, rather than the average. It provides better information and it looks like there is room to fit these in.

360-365: This would seem to argue towards using only (or primarily) the repeat orbits for the S2S instead of all overlaps, right? Or at least limiting to overlapping orbits that have orientations that limit the geo-location error effect?

405: I would note though that more advanced techniques, such as Backus-

Gilbert, do take account of the antenna pattern of the sensor and the measurement response function (MRF). So, it should be better than simple interpolation. Another approach that uses MRF for weighting is Brodzik et al., https://doi.org/10.5067/MEASURES/CRYOSPHERE/NSIDC-0630.001.

418: minor grammar suggestion, "...often leads to increases in the noise level."

Figure 9: Why does there appear to be more vectors on the Atlantic side of the Arctic than the Pacific side? I would expect the pattern to essentially be symmetric, but in the East Greenland, Barents, and Kara Seas, there are more vectors than at the same latitudes in the Beaufort, Chukchi, East Siberian, and Laptev Seas.

444: typo, "...larger than discussed here..."

---

## Author Comment (AC3) · 10 Feb 2021

Summary

This paper presents an approach to calculate swath-to-swath (S2S) sea ice motion vectors from passive microwave imagery. Via comparisons with buoys, this approach is shown to be more accurate that the standard daily map products that composite brightness temperatures over a 24-hour period. The S2S are improved because the TB values are instantaneous instead of a "blurred" average and the time between images is exact as opposed an average time of all passes, which also "blurs" the motion estimates. The methodology is promising for the future CIMR mission, which will have wider swaths to obtain more motion vectors and higher spatial resolution for greater accuracy.

General Comment

Swath-to-swath sea ice motion vectors has been a long-discussed idea, so it is great to see it successfully implemented here. The methodology is sound and it appears to be viable to do operationally. The paper is well-written and the results are convincing.

*We thank the reviewer for his positive comments.*

I have a couple general comments. First, there is discussion of the packaging of the fields in Section 5, but it's still not totally clear how this would happen. I can understand the baseline approach, where there are fields for each of the overlaps and the times. But also discussed is the generation of daily maps. I agree that this would be useful and would probably be most convenient for most users. But it's not clear how this would be created. You would have motions for time periods of ∼100 minutes to over 24 hours. How would the different time separations be combined? The ice motion will vary between the periods, so simple interpolation/extrapolation may not work. The easiest thing would be to use the repeat orbit overlaps from each day – then there would be 24 hours between all vectors. But, of course, this leaves out many vectors. Maybe there could be some kind of weighting scheme to optimally combine vectors over different time intervals into an optimum cohesive daily map.

*These are very exciting thoughts and we might be investigating such merging methodologies in future studies. We do not feel confident expanding on this aspects in the present manuscript though. Still, we modified two sentences in the Discussions section to bring some elements of your comment:*

*As with the other parameters to be observed by CIMR, a Level-3 sea-ice drift product should be prepared that optimally combines e.g. a day's worth of S2S products (having very different time durations) into a complete map of e.g. ~24 hours sea-ice motion vectors. To the best of our knowledge, such merging algorithms do not exist.*

Another thing that came to mind while reading is the potential utility for summer motions. It's understandable to focus first on winter, but summer is not mentioned until Section 5. There are well-known limitations to using PM TBs for summer – most notably the surface melt and (especially for 89 GHz) greater water vapor levels. The S2S approach seems like it would be potentially quite helpful. First, the exact time of S2S will remove some error because ice is more dynamic in the summer, and potentially improve accuracy of the more sensitive lower frequency channels. For example, 18.7 GHz S2S may obtain better summer motions than daily composites. On the other hand, I wonder if the instantaneous S2S fields might cause some problems for the 89 GHz channels because water vapor can change rapidly and the "smearing" of the daily composite TBs may filter out some of that variability that could cause errors. It would be great to have a summer example in the paper, but I think keeping the focus tighter

on the winter case studies makes sense here. But I think some brief discussion of the limitations of PM for summer motions, e.g., in the Introduction, and a little more discussion in Section 5, would be helpful.

*Since the other reviewer (and editor) also pointed out that an investigation of the summer conditions would strengthen the paper, we decided to include this in revising the manuscript. As already mentioned in the manuscript, a major limitation to today's ice motion retrieval during summer is the relatively coarse resolution of the 18.7 GHz imagery channels of the AMSR2 sensor. Adopting an S2S versus DM approach might help, but the step-change will be the spatial resolution at CIMR, which we cannot test at present. This is even more true for lower frequencies (10.6 GHz or 6.9 GHz). We will include an analysis of summer sea-ice drift from AMSR2 18.7 GHz (DM vs S2S) in the revision of our manuscript.*

A few other minor comments are noted below. These are addressable in my view with minor revisions.

Specific Comments (by line number):
43: "short-lived" is ambiguous here. It may suggest something that lasts only a few weeks, but buoys can last at least a few years. That's short compared to long-term climate monitoring, but longer than what I would call "short-lived".

*Agreed, see below.*

44: "scattered with vast distances between them" can be described more simply as "sparse". Perhaps rephrase this whole sentence to something more like: "Buoys have a limited lifespan before they exit out of the Arctic or the ice melts; this and limited opportunities for deployment result in sparse spatial coverage of the Arctic."

*We follow your suggestion and revise as: "Nevertheless, buoys have a limited lifespan before the sea-ice floe they seat on melts, or they drift out of the Arctic, or they suffer technical issues; this and limited opportunities for deployment result in sparse spatial coverage."*

76-77: I don't see a reason to abbreviate "Section" here – it's more readable without the abbreviation.

*The author's guidelines from EGU TC read: The abbreviation "Sect." should be used when it appears in running text and should be followed by a number unless it comes at the beginning of a sentence.*

81: Is there a specific citation recommended? At the least the website should be given, but if at all possible a formal citation (with date of access for a website) should be used.

*This was the Global Portal System (G-portal) (*https://gportal.jaxa.jp/gpr/*), and we now specify it in the text and the acknowledgments section.*

Table 1: I would recommend giving the diameters of both dimensions of the sensor footprints, e.g., A x B, rather than the average. It provides better information and it looks like there is room to fit these in.

*Showing the two diameters (A x B) would also be our preferred solution. However, because CIMR is still in the design phase, these numbers are not known at this stage. For CIMR, we must thus*

*keep these requirements on the arithmetic mean. We suggest to add (AxB) for AMSR2. We modify Table 1 and its caption:*

| Band | L | C | X | Ku | Ka | W |
|---|---|---|---|---|---|---|
| Center Frequency [GHz] | 1.4 | 6.9 | 10.7 | 18.7 | 36.5 | 89.0 |
| AMSR2 -[km] | - | 49 (35 x 62) | 33 (24 x 42) | 18 (14 x 22) | 9 (7 x 12) | 4 (3 x 5) |
| CIMR -[km] | <60 | 15 | 15 | 5 | <5 | - |

Table 1: Spatial resolution ( arithmetic mean of the minor and major diameters of the instantaneous field-of-view ellipse, and the two diameters for AMSR2) of selected microwave frequencies of the AMSR2 and CIMR missions. AMSR2 also records at 7.3 and a 23.8 GHz, those will not be on-board CIMR. "-" indicates a microwave frequency is not recorded by the mission. The values for CIMR are the mission requirements from Donlon et al. (2020), those for AMSR2 are from the Observing Systems Capability Analysis and Review (OSCAR) tool of the World Meteorological Organization. The CIMR mission being under development, the actual diameters of the ellipses are not known at time of writing. See also Lavergne (2018) for a graphical representation of these values.

360-365: This would seem to argue towards using only (or primarily) the repeat orbits for the S2S instead of all overlaps, right? Or at least limiting to overlapping orbits that have orientations that limit the geo-location error effect?
*Indeed, which is why we noted in the text that "the retrieval of accurate sea-ice drift vectors from individual swaths puts stringent requirements on the geo-location accuracy of passive microwave missions **if all swath overlap pairs should be processed**".*

*We do not think it is necessary to bring more information at this point, since we so far only explored a very specific type of geo-location error (systematic offset in flight direction) while other types of geo-location errors (e.g. along the scan direction) will result in other characteristics of the bias.*

405: I would note though that more advanced techniques, such as Backus-Gilbert, do take account of the antenna pattern of the sensor and the measurement response function (MRF). So, it should be better than simple interpolation. Another approach that uses MRF for weighting is Brodzik et al., https://doi.org/10.5067/MEASURES/CRYOSPHERE/NSIDC-0630.001.

*Yes, we re-formulated as "However, more advanced gridding techniques (e.g. Backus-Gilbert) could also be challenged by the lack of sufficient overlap between neighboring 89 GHz Field of Views".*

418: minor grammar suggestion, ". . .often leads to increases in the noise level."

*Implemented.*

Figure 9: Why does there appear to be more vectors on the Atlantic side of the Arctic than the Pacific side? I would expect the pattern to essentially be symmetric, but in the East Greenland, Barents, and Kara Seas, there are more vectors than at the same latitudes in the Beaufort, Chukchi, East Siberian, and Laptev Seas.

*This is a good question that we had to investigate in more details. The larger amount of S2S vectors in the "European" sector of the Arctic is a consequence of the orbit cycle of CIMR. The last orbit to*

*start in a 48 hours period [D@00utc to D+2@00:00 utc] also extends in the following day, and leads to a region of the Arctic (and Antarctic) to be covered by one more swath. This results into additional swath overlaps, and thus S2S vectors. Furthermore, the feature is not fixed in space, and will transit all around the pole in the 29 days/412 orbit cycle of CIMR.*

*When revising the paper, we will re-assess Fig. 9 and the way we count the last orbit in the 48 hours period. This might lead us to re-doing Fig. 9 and obtain a more symmetrical coverage. We will in any case comment in greater details the link to the orbit cycle.*

444: typo, ". . .larger than discussed here. . ."

*Implemented.*

---

## Author Response (AR1)

This document contains our answers to two reviewers, also noting the edits implemented in the manuscript to accommodate the suggestions. Our answers are marked with a purple color.

**===== REVIEWER 1 ======**

Conventionally, sea ice motion from passive microwave observations is extracted from aggregated brightness temperature daily products covering the entire Arctic or Antarctic domains. This paper investigates the possibility of deriving sea ice motion vectors directly from the overlapping AMSR2 individual swaths (S2S scenario) as opposed to the daily products (DM scenario) and implications on the future ESA CIMR mission. A well-established ice motion tracking algorithm based on the Continuous Maximum Cross-Correlation (CMCC) approach was applied to derive ice motion vectors in both the S2S and DM scenarios. The authors demonstrated that a much larger number of ice motion vectors with higher accuracy (as validated against in-situ buoys) is derived in the case of S2S compared to DM scenario. The S2S ice motion extraction scenario is recommended to be applied to the future CIMR mission, which will provide a higher spatial resolution compared to AMSR2. This is an interesting paper, but I have the following comments which need to be addressed before the manuscript can be considered for publication.

**We thank the reviewer for his/her comments and provide some elements of answers below.**

**Major comments:**

1. In this study, the authors used only winter time periods for both the Arctic and Antarctic. What about the summer time? Could S2S approach provide better (or any reasonable) ice motion tracking results compared to the DM approach in summer time? Would lower remote sensing frequencies be recommended in that case (due to the larger penetration depth) as opposed to the higher frequencies? I think the paper will look much better if quantitative evaluation of ice motion provided by S2S versus DM during the summer time is presented.

We include an analysis of summer sea-ice drift from AMSR2 36.5 GHz (DM vs S2S) in the revision of our manuscript (new section 4.4, new Fig. 6, see below). In the summer season, S2S and DM mostly perform the same (slightly better with DM) and the most important processing aspect is to compute motion vectors over short time durations (18 h or 24 h rather than 48 h). Covering a longer time period introduced some edits in section 2. Data.

4.4 Seasonal evolution of the drift accuracy in the Arctic

The two last sections focused on two 3 months winter periods in the Arctic and Antarctic. Here, we present monthly validation results covering Oct 2019 to Dec 2020 (15 months) in the Arctic. Our main objective is to investigate if the S2S approach helps the retrieval of sea-ice drift vectors during the Arctic summer melt season. Due to surface melt and increased wetness in the atmosphere, the tracking of sea-ice drift from passive microwave instruments has traditionally been a challenge during summer. While Kwok (2008) has shown that imagery from the AMSR2 mission can be used to track summer sea-ice drift (using a DM approach), the accuracy when compared to buoy trajectories was shown to be much reduced.

During summer in the Arctic, the atmosphere is wetter and contributes significantly to the brightness temperature recorded at 36.5 GHz, effectively hiding more of the surface emissivity. The surface emissivity is also more variable in time because of the cycles of sub-daily cycles of melt/freezing (early and late in the summer season) and the direct impact of weather system traveling over the sea-ice. It is thus not a surprise to see better validation statistics with shorter than longer drift durations since a shorter duration will increase the chance of tracking the same surface emissivity patterns with less chances for a change happening in the time between the two images.

Fig. 6 shows monthly validation statistics for several DM and S2S products obtained from the AMSR2 36.5 GHz

Figure 6: a) Monthly validation statistics of the S2S and DM drift vectors with drift durations 48 h (blue), 24 h (orange) and 18 h (S2S only) from Oct 2019 to Dec 2020 in NH. b) Number of collocation matchups per months for the DM products (black: total, blue: Ice Tethered Profilers, and orange: seaiceportal.de). Both RMSE and BIAS are reported. The summer season (May-Sept) is greyed.

imagery. Both 48 h, 24 h and 18 h drift products were prepared and validated following Sect. 3.3. Fig. 6 confirms that the validation statistics of drift vectors with shorter durations (e.g. 18 h and 24 h) are better than those of vectors with longer durations (48 h), both in terms of RMSE and bias, and for the whole 15 months period. This was already noted in Sect. 4.3 for the period Oct-Dec 2019. Fig. 6 also confirms that, for most of the year, S2S drift vectors reach better validation statistics than DM vectors. This is true for all the winter months (Oct – Apr). However, the better accuracy of S2S drift vectors is not apparent during the summer months (May – Sept) when DM reaches (slightly) better results. Validation results during summer are indeed worse than during winter, but the main driver for the worsen accuracy in summer seems to be the duration of the drift vectors (24 h vs 48 h), not the adoption of an S2S vs a DM approach.

When conducting the same investigations with the 18.7 GHz imagery of AMSR2 (not shown) we found roughly the same results but the validation statistics were slightly worse than those obtained with 36.5 GHz throughout the year. The 18.7 GHz microwave frequency is emitted from deeper in the sea ice and snow medium and is less affected by the atmosphere, so that one would expect more stable surface emissivity patterns available for sea-ice motion tracking (Kwok, 2008). However the coarser resolution of the 18.7 GHz frequency channels (Table 1) works against this property by blurring the emissivity patterns.

We note that, even if DM vectors seem to validate better than S2S vectors during the summer melt season, adopting the S2S approach still gives many more vectors per day than the DM approach.

2. The authors discuss the differences in sea ice motion tracking from different frequency channels (mainly Ka and W due to their relatively high spatial resolution). However, polarization options were not discussed. What are the differences in terms of the number and accuracy of ice motion vectors derived from the horizontal and vertical polarization swaths? What optimum polarization option or polarization combinations are recommended for the ice motion tracking?

These are good questions, and we now see that we have not included some relevant elements of the motion tracking methodology in section 3.1. We indeed already use both vertically and horizontally polarized imagery channels, as described in Lavergne et al. (2010) section 2.3 "Combining Several Imaging Channels". In short, for each microwave frequency, we combine the information content of the vertically and horizontally polarized imagery by finding the maximum of the sum of the cross-correlation from each polarization independently. This retrieves a single motion vector from two polarization channels.

*We add the following text to section 3.1:*

Second, for a given microwave frequency, the information content of both the vertically and horizontally polarized images are combined within the optimization of the cross-correlation function. In practice, and following Lavergne et al. (2010), the solution sea-ice drift vectors are at the maximum of the sum of two cross-correlation functions : one from of the vertically polarized imagery, and one from the horizontally polarized imagery. The reader is referred to the discussion in Lavergne et al. (2010, section 3.2) for a description of this approach. In the remaining of our paper, despite mentioning only the microwave frequency, we do use both polarizations in the motion tracking.

**Technical corrections:**

There is some language inaccuracies in the paper. I tried to point out some of them below with suggested changes.

Consider to mark figure panels with letters (a), (b), etc.

This was implemented for several figures.

Line 37. "These can...". It seems that some word between "These" and "can" is missing.

The missing words were "on-ice buoys".

Line 194. "over a Northern and a Southern Hemisphere grid.". Should "a" be replaced with "the"?

**Indeed, this was changed.**

Line 232. "A first" => "The first". Line 234. "...very different characteristics to the DM products..." => "...very different characteristics compared to the DM products..."

**Both implemented.**

Line 242. "...these mean times associated with the DM ice drift product are averaged values..." => "...these mean times associated with the fact that the DM ice drift product are averaged values..."

We revised our sentence to read : "... these mean times associated with the DM ice drift vectors are values averaged over several overlapping swaths... "

Line 278. "...the low number of validation data..." => "...the lower number of validation data points..."

Implemented.

Line 284. "...but this time studying..." => "...but this time we consider..."

**Implemented.**

Line 285 and 289, and throughout the text. "100 mn => "100 min".

**Implemented throughout the text.**

Line 318. "Fig. 6 is a repeat of Fig. 5...". In fact, Fig. 6 is similar to Fig. 5 (left, NH) and not the entire Fig. 5. Please reflect it accordingly in the text.

Indeed, this was implemented.

Fig.7 and Fig.8. Please move the figure title to the figure caption.

Thank you, this will be implemented.

===== REVIEWER 2 ======

**Summary**

This paper presents an approach to calculate swath-to-swath (S2S) sea ice motion vectors from passive microwave imagery. Via comparisons with buoys, this approach is shown to be more accurate that the standard daily map products that composite brightness temperatures over a 24-hour period. The S2S are improved because the TB values are instantaneous instead of a "blurred" average and the time between images is exact as opposed an average time of all passes, which also "blurs" the motion estimates. The methodology is promising for the future CIMR mission, which will have wider swaths to obtain more motion vectors and higher spatial resolution for greater accuracy.

**General Comment**

Swath-to-swath sea ice motion vectors has been a long-discussed idea, so it is great to see it successfully implemented here. The methodology is sound and it appears to be viable to do operationally. The paper is well-written and the results are convincing.

**We thank the reviewer for his positive comments.**

I have a couple general comments. First, there is discussion of the packaging of the fields in Section 5, but it's still not totally clear how this would happen. I can understand the baseline approach, where there are fields for each of the overlaps and the times. But also discussed is the generation of daily maps. I agree that this would be useful and would probably be most convenient for most users. But it's not clear how this would be created. You would have motions for time periods of  $\sim$ 100 minutes to over 24 hours. How would the different time separations be combined? The ice motion will vary between the periods, so simple interpolation/extrapolation may not work. The easiest thing would be to use the repeat orbit overlaps from each day – then there would be 24 hours between all vectors. But, of course, this leaves out many vectors. Maybe there could be some kind of weighting scheme to optimally combine vectors over different time intervals into an optimum cohesive daily map.

These are very exciting thoughts and we might be investigating such merging methodologies in future studies. We do not feel confident expanding on this aspects in the present manuscript though.

Still, we modified two sentences in the Discussions section to bring some elements of your comment:

As with the other parameters to be observed by CIMR, a Level-3 sea-ice drift product should be prepared that optimally combines e.g. a day's worth of S2S products (having very different time durations) into a complete map of e.g. ~24 hours sea-ice motion vectors. To the best of our knowledge, such merging algorithms do not exist.

Another thing that came to mind while reading is the potential utility for summer motions. It's understandable to focus first on winter, but summer is not mentioned until Section 5. There are well-known limitations to using PM TBs for summer – most notably the surface melt and (especially for 89 GHz) greater water vapor levels. The S2S approach seems like it would be potentially quite helpful. First, the exact time of S2S will remove some error because ice is more dynamic in the summer, and potentially improve accuracy of the more sensitive lower frequency channels. For example, 18.7 GHz S2S may obtain better summer motions than daily composites. On the other hand, I wonder if the instantaneous S2S fields might cause some problems for the 89 GHz channels because water vapor can change rapidly and the "smearing" of the daily composite TBs may filter out some of that variability that could cause errors. It would be great to have a summer example in the paper, but I think keeping the focus tighter on the winter case studies makes sense here. But I think some brief discussion of the limitations of PM for summer motions, e.g., in the Introduction, and a little more discussion in Section 5, would be helpful.

We include an analysis of summer sea-ice drift from AMSR2 36.5 GHz (DM vs S2S) in the revision of our manuscript (new section 4.4, new Fig. 6, see below). In the summer season, S2S and DM mostly perform the same (slightly better with DM) and the most important processing aspect is to compute motion vectors over short time durations (18 h or 24 h rather than 48 h). Covering a longer time period introduced some edits in section 2. Data.

4.4 Seasonal evolution of the drift accuracy in the Arctic

The two last sections focused on two 3 months winter periods in the Arctic and Antarctic. Here, we present monthly validation results covering Oct 2019 to Dec 2020 (15 months) in the Arctic. Our main objective is to investigate if the S2S approach helps the retrieval of sea-ice drift vectors during the Arctic summer melt season. Due to surface melt and increased wetness in the atmosphere, the tracking of sea-ice drift from passive microwave instruments has traditionally been a challenge during summer. While Kwok (2008) has shown that imagery from the AMSR2 mission can be used to track summer sea-ice drift (using a DM approach), the accuracy when compared to buoy trajectories was shown to be much reduced.

Fig. 6 shows monthly validation statistics for several DM and S2S products obtained from the AMSR2 36.5 GHz imagery. Both 48 h, 24 h and 18 h drift products were prepared and validated following Sect. 3.3. Fig. 6 confirms that the validation statistics of drift vectors with shorter durations (e.g. 18 h and 24 h) are better than those of vectors with longer durations (48 h), both in terms of RMSE and bias, and for the whole 15 months period. This was already noted in Sect. 4.3 for the period Oct-Dec 2019. Fig. 6 also confirms that, for most of the year, S2S drift vectors reach better validation statistics than DM vectors. This is true for all the winter months (Oct – Apr). However, the better accuracy of S2S drift vectors is not apparent during the summer months (May – Sept) when DM reaches (slightly) better results. Validation results during summer are indeed worse than during winter, but the main driver for the worsen accuracy in summer seems to be the duration of the drift vectors (24 h vs 48 h), not the adoption of an S2S vs a DM approach.

During summer in the Arctic, the atmosphere is wetter and contributes significantly to the brightness temperature recorded at 36.5 GHz, effectively hiding more of the surface emissivity. The surface emissivity is also more variable in time because of the cycles of sub-daily cycles of melt/freezing (early and late in the summer season) and the direct impact of weather system traveling over the sea-ice. It is thus not a surprise to see better validation statistics with

shorter than longer drift durations since a shorter duration will increase the chance of tracking the same surface emissivity patterns with less chances for a change happening in the time between the two images.

When conducting the same investigations with the 18.7 GHz imagery of AMSR2 (not shown) we found roughly the same results but the validation statistics were slightly worse than those obtained with 36.5 GHz throughout the year. The 18.7 GHz microwave frequency is emitted from deeper in the sea ice and snow medium and is less affected by the atmosphere, so that one would expect more stable surface emissivity patterns available for sea-ice motion tracking (Kwok, 2008). However the coarser resolution of the 18.7 GHz frequency channels (Table 1) works against this property by blurring the emissivity patterns.